# Bmp8a is an essential positive regulator of antiviral immunity in zebrafish

Shenjie Zhong [1,2], Haoyi Li[1,2], Yun-Sheng Wang[1,2], Ying Wang[1,2], Guangdong Ji[1,2], Hong-Yan Li [1,2], Shicui Zhang [1,2✉] & Zhenhui Liu [1,2✉]

Bone morphogenetic protein (BMP) is a kind of classical multi-functional growth factor that plays a vital role in the formation and maintenance of bone, cartilage, muscle, blood vessels, and the regulation of adipogenesis and thermogenesis. However, understanding of the role of BMPs in antiviral immunity is still limited. Here we demonstrate that Bmp8a is a newly-identified positive regulator for antiviral immune responses. The *bmp8a*[−/−] zebrafish, when infected with viruses, show reduced antiviral immunity and increased viral load and mortality. We also show for the first time that Bmp8a interacts with Alk6a, which promotes the phosphorylation of Tbk1 and Irf3 through p38 MAPK pathway, and induces the production of type I interferons (IFNs) in response to viral infection. Our study uncovers a previously unrecognized role of Bmp8a in regulation of antiviral immune responses and provides a target for controlling viral infection.

[1] College of Marine Life Science and Institute of Evolution & Marine Biodiversity, Ocean University of China, Qingdao, China. [2] Laboratory for Marine Biology and Biotechnology, Pilot National Laboratory for Marine Science and Technology (Qingdao), Qingdao, China. ✉email: sczhang@ouc.edu.cn; zhenhuiliu@ouc.edu.cn

Viruses infect all groups of living things and produce a variety of diseases. Host cell detection of viral infection depends on the recognition of pathogen-associated molecular patterns (PAMPs) from viral proteins or nucleic acids by the pattern recognition receptors (PRRs) such as Toll-like receptors (TLRs), RIG-I-like receptors (RLRs), NOD-like receptors (NLRs), and cytoplasmic DNA or RNA sensors[1–6]. The PRRs, once stimulated by their appropriate ligands, trigger distinct intracellular signaling pathways that converge on the activation of IFN regulatory factor 3 (IRF3) and/or IFN regulatory factor 7 (IRF7), which then translocate into the nucleus and activate the production of interferons (IFNs) that play important roles in inhibiting virus replication[7,8].

IFNs are classified into type I IFNs that include IFNα and IFNβ; type II IFN which has only a single member IFN-γ; and type III IFNs that consist of IFN-λs[9]. All these three types of IFNs trigger intracellular signaling cascades generally via the Janus kinase signal transducer and activator of transcription (JAK-STAT) pathway[10]. Canonically, type I IFNs initiate the signaling via binding to a heterodimeric receptor complex IFN-α/β receptor (IFNAR), inducing phosphorylation of transcription factors STAT1 and STAT2. The phosphorylated STATs form a trimolecular complex with IRF9, which translocates to the nucleus and binds to IFN-stimulated response elements (ISREs) of IFN-stimulated genes (ISGs), resulting in transcriptional activation of the target ISGs, including those encoding numerous cytokines and antiviral proteins[11–16]. In contrast, type II IFN binds to its unique receptor, IFN-γ receptor (IFNGR), resulting in the phosphorylation of STAT1 and the formation of STAT1 homodimers that recognize gamma-activated sequences (GASs) present in the promoter regions of IFN-γ-regulated genes[9].

Bone morphogenetic proteins (BMPs) are potent growth factors belonging to the transforming growth factor beta (TGFβ) superfamily, which are intercellular signaling molecules with multiple functions in development and differentiation, such as skeletal formation and hematopoiesis, as well as lipid oxidation[17–21]. However, accumulating data suggests that BMPs also play an important roles in the regulation of immune responses[22–27]. For example, BMP7 has been shown to drive Langerhans cell differentiation[28]. BMP6 has anti-HCV activity independently of IFN, but can also enhance the response to IFN[29]. Interestingly, BMP signaling appears to have different responses in T cells: BMP4 and BMP6 were shown to promote T cell proliferation, whereas BMP2 was seen to inhibit T cell proliferation[30,31]. In addition, although BMPs were found to promote tumor growth by inhibiting the functions of cytotoxic T lymphocytes (CTLs) and dendritic cells (DCs), they were also shown to inhibit cancer cell proliferation and promote the activity of NK cells that typically display antitumour activity[32]. Therefore, it remains controversial regarding the functions of BMPs in immune responses.

In humans and mice, two closely related BMP8 genes, BMP8A and BMP8B, have been identified[33,34]. BMP8B has been found to play roles in the regulation of thermogenesis in mature brown adipose tissue and spermatogenesis in male germ cells, while BMP8A shown to closely correlate with the progression of spermatogenesis[35,36]. In contrast, only one bmp8 gene, bmp8a, has been isolated in zebrafish, and shown to be involved in the regulation of lipid metabolism[37]. To date, whether Bmp8a plays a role in immune responses is completely unknown. The present study was thus performed to answer this question. We demonstrated that Bmp8a was a previously unrecognized regulator functioning in antiviral immune responses of zebrafish *Danio rerio*, providing a target for control of viral infection.

## Results

**Bmp8a inhibited RNA virus replication in vitro**. To explore the function of zebrafish Bmp8a in the antiviral immune response, we infected both the wild-type zebrafish liver (ZFL) cells and ZFL cells over-expressing bmp8a as well as the epithelioma papulosum cyprini (EPC) cells and EPC cells over-expressing bmp8a with Grass carp reovirus (GCRV), a dsRNA virus, and then monitored the cytopathic effect (CPE) and the viral titers of the supernatants. In response to GCRV challenge, an apparent CPE was observed in the control cells, while the bmp8a-overexpressing ZFL cells had much less CPE (Fig. 1a). In accordance with the CPE results, a significant decrease of the viral titers was observed in the supernatants of bmp8a-overexpressing ZFL cells (Fig. 1b). Similarly, the GCRV-induced CPE and GCRV yields of the supernatants were also markedly reduced in the bmp8a-overexpressing EPC cells (Fig. 1c, d). Moreover, the recombinant Bmp8a protein also blocked GCRV replication in ZFL cells (Fig. 1e, f) or EPC cells (Fig. 1g, h). These suggested that Bmp8a suppressed the replication of the RNA virus in the cells.

**Bmp8a was involved in antiviral responses in vivo**. To test the function of Bmp8a in host defense against virus infection in vivo, we generated *bmp8a* deficient (*bmp8a*$^{-/-}$) zebrafish using the TALEN approach (Supplementary Fig. 1a), which caused seven nucleotides deletion in the exon 4 (Supplementary Fig. 1b, c). Although BMPs are pleiotropic, the zebrafish *bmp8a* knockout did not result in lethality. In fact, the mutant lines (*bmp8a*$^{-/-}$) can survive and breed normally under standard laboratory conditions. We challenged both the wild-type and *bmp8a*$^{-/-}$ mutant zebrafish by intraperitoneal injection with GCRV. Compared with wild-type zebrafish, the levels of GCRV RNA in the liver, kidney, intestine, and spleen of *bmp8a*$^{-/-}$ zebrafish were considerably increased (Fig. 1i). Accordingly, *bmp8a*$^{-/-}$ zebrafish exhibited significantly reduced survival rate than wild-type zebrafish upon GCRV infection (Fig. 1j). When the wild-type and *bmp8a*$^{-/-}$ zebrafish were similarly challenged with spring viremia of carp virus (SVCV), a negative ssRNA virus, the survival rate of *bmp8a*$^{-/-}$ zebrafish was also remarkably lower than that of wild-type zebrafish (Fig. 1k). Moreover, when the wild-type and *bmp8a*$^{-/-}$ zebrafish were infected with turbot skin verruca disease virus (TSVDV) isolated from skin (Supplementary Fig. 2), death occurred in *bmp8a*$^{-/-}$ zebrafish at 108 h of post-infection, and all the fish died at 156 h of post-infection, while none of the wild type zebrafish were dead during the period (Fig. 1l). These data indicated that *bmp8a*$^{-/-}$ zebrafish were more susceptible to virus infection than wild-type zebrafish, suggesting involvement of Bmp8a in antiviral immune responses in vivo.

**Bmp8a promoted expression of antiviral genes**. In zebrafish, type I IFNs contained 4 members *ifnφ1*, *ifnφ2*, *ifnφ3*, and *ifnφ4*, among which *ifnφ1* and *ifnφ3* were shown to be involved in the antiviral response[38]. In contrast, in carp, only one type I IFN was identified to fulfill antiviral role in EPC cell[39,40]. To pinpoint the role of Bmp8a in the antiviral immune response, both the wild-type and *bmp8a*$^{-/-}$ zebrafish were injected intraperitoneally with GCRV, and the expression of type I IFN genes and the antiviral protein gene *mx* were detected using quantitative reverse transcription PCR (qRT-PCR). We found that the expression of both *ifnφ1* and *ifnφ3* as well as *mx* was all significantly downregulated in the liver, kidney, intestine, and spleen of *bmp8a*$^{-/-}$ zebrafish, compared with those of wild-type fish (Fig. 2a–d). We then overexpressed bmp8a in ZFL cells and found that ZFL cells with overexpressed Bmp8a showed markedly higher expression of *ifnφ1*, *ifnφ3*, and *mx* than that of control cells, infected with or without GCRV (Fig. 2e–h). In addition, *bmp8a* knockdown

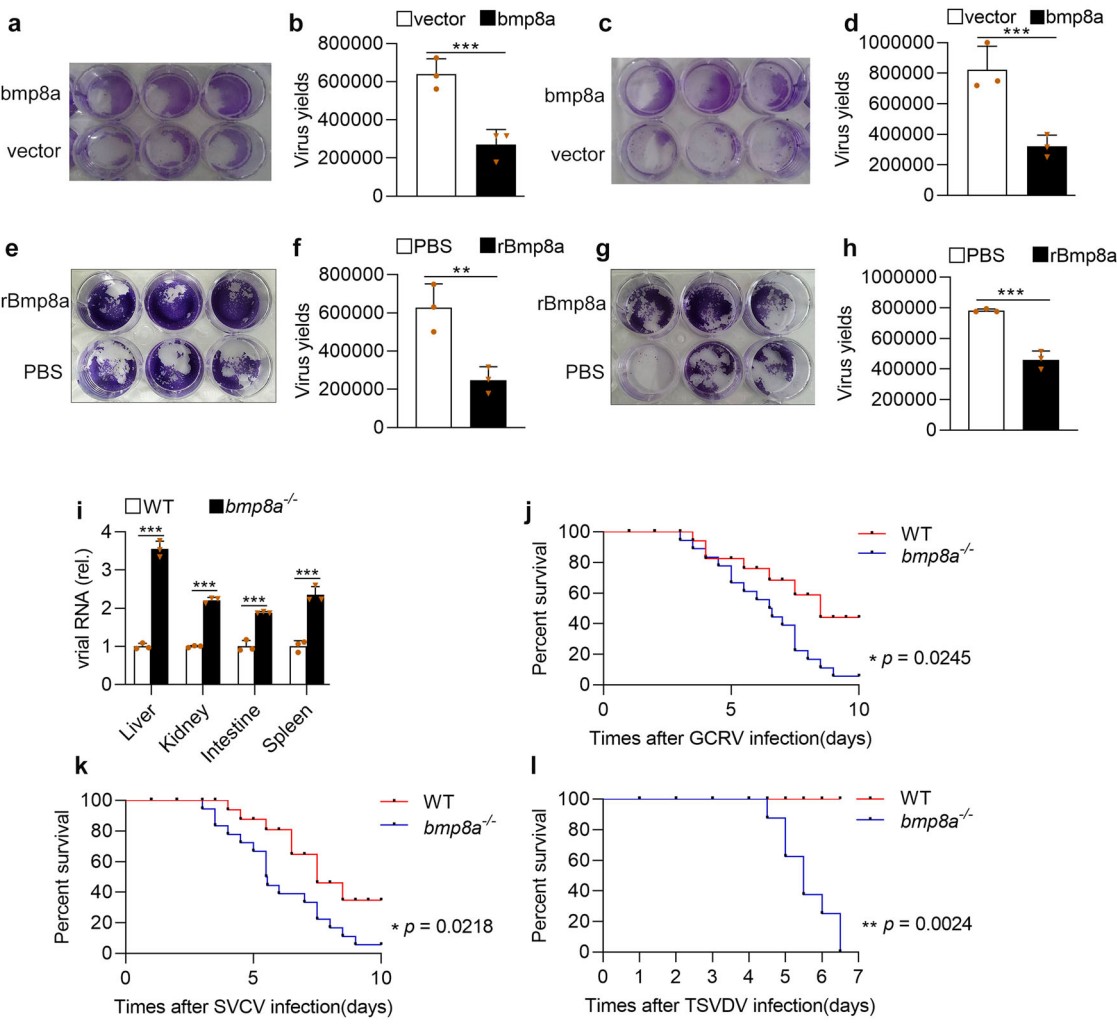

**Fig. 1 Bmp8a inhibits RNA virus replication both in vitro and in vivo. a**, **b** ZFL cells were transfected with bmp8a (1 μg) or empty vector (1 μg), respectively. The cells were infected with GCRV ($5 \times 10^4$ TCID$_{50}$ per ml) at 24 h of post-transfection, and the culture supernatants were collected at 72 h of post-infection. The cell monolayers were fixed with 4% paraformaldehyde for 1 h and stained with 0.5% crystal violet for 2 h (**a**), and the viral titers of the supernatants were determined by TCID$_{50}$ assays (**b**). **c**, **d** Similar as (**a**, **b**) but in EPC cells. **e**, **f** ZFL cells were treated with rBmp8a (final concentrations of 5 μg/ml) or PBS co-incubation with GCRV ($5 \times 10^4$ TCID$_{50}$ per ml), and the culture supernatants were collected at 72 h of post-infection. The cell monolayers were fixed with 4% PFA for 1 h and stained with 0.5% crystal violet for 2 h (**e**), and the viral titers of the supernatants were determined by TCID$_{50}$ assays (**f**). **g**, **h** Similar as (**e**, **f**) but in EPC cells. **i** The expression of GCRV RNA in the liver, kidney, intestine, and spleen from wild-type (WT) or bmp8a$^{-/-}$ zebrafish injected i.p. with 50 μl of GCRV ($10^8$ TCID$_{50}$ per ml) for 72 h. **j**, **k** Kaplan–Meier analysis of the overall survival of WT (n = 20) or bmp8a$^{-/-}$ zebrafish (n = 20) which were injected i.p. with 50 μl of GCRV ($10^8$ TCID$_{50}$ per ml) or SVCV ($10^8$ TCID$_{50}$ per ml) and monitored every 12 h after infection. **l** Kaplan–Meier analysis of the overall survival of WT (n = 8) or bmp8a$^{-/-}$ zebrafish (n = 8) which were injected i.p. with 50 μl of TSVDV (crude virus extracts filtered by a 0.45 μm microporous membrane) and monitored every 12 h after infection. The expression of zebrafish actb1 was used as an internal control for the qRT-PCR. Data were from three independent experiments (**a-i**) or two independent experiments (**j-l**). Data were analyzed by Student's t-test (two-tailed) or log-rank (Mantel–Cox) test and were presented as mean ± SD (*p < 0.05, **p < 0.01, ***p < 0.001).

caused remarkably decreased expression of *ifnφ1*, *ifnφ3*, and *mx* in ZFL cells (Fig. 2i–l). Similarly, the expression of *ifn* and *mx* in EPC cells with overexpressed Bmp8a was also remarkably upregulated than that in control cells (Fig. 2m–p). Moreover, EPC cells cotransfected with bmp8a expressing plasmid and IFNφ1, IFNφ3, or EPC IFN promoter-driven luciferase plasmid, followed by infection with or without GCRV, had markedly increased intracellular IFNφ1, IFNφ3, or EPC IFN promoter-driven luciferase activities (Fig. 2q–s). All these data denoted that Bmp8a promoted the expression of both type I IFN and *mx* genes.

**Bmp8a activated Tbk1–Irf3–Ifn antiviral signaling via p38 MAPK pathway.** To putatively examine the molecular

mechanism by which Bmp8a stimulates *ifn* and *mx* expression, considering that the Tbk1–Irf3/7 signaling pathway is crucial in the activation of *ifn* transcription, we detected the expression and phosphorylation of the Tbk1 and Irf3 upon overexpressing or knockdown of *bmp8a*. We found that the expressions of *tbk1*, *irf3*, and *irf7* were significantly upregulated in bmp8a-overexpressing ZFL cells than control cells, infected with or without GCRV (Fig. 3a–d). Similar expression patterns of *tbk1*, *irf3*, and *irf7* were also observed in EPC cells, infected with or without GCRV (Fig. 3e–h). In addition, *bmp8a* knockdown resulted in considerably reduced expression of *tbk1*, *irf3*, and *irf7* in ZFL cells than control cells, infected with or without GCRV (Fig. 3i–l). We then evaluated the activation of Tbk1 and Irf3 by immunoblot assays in both bmp8a-overexpressing and wild-type

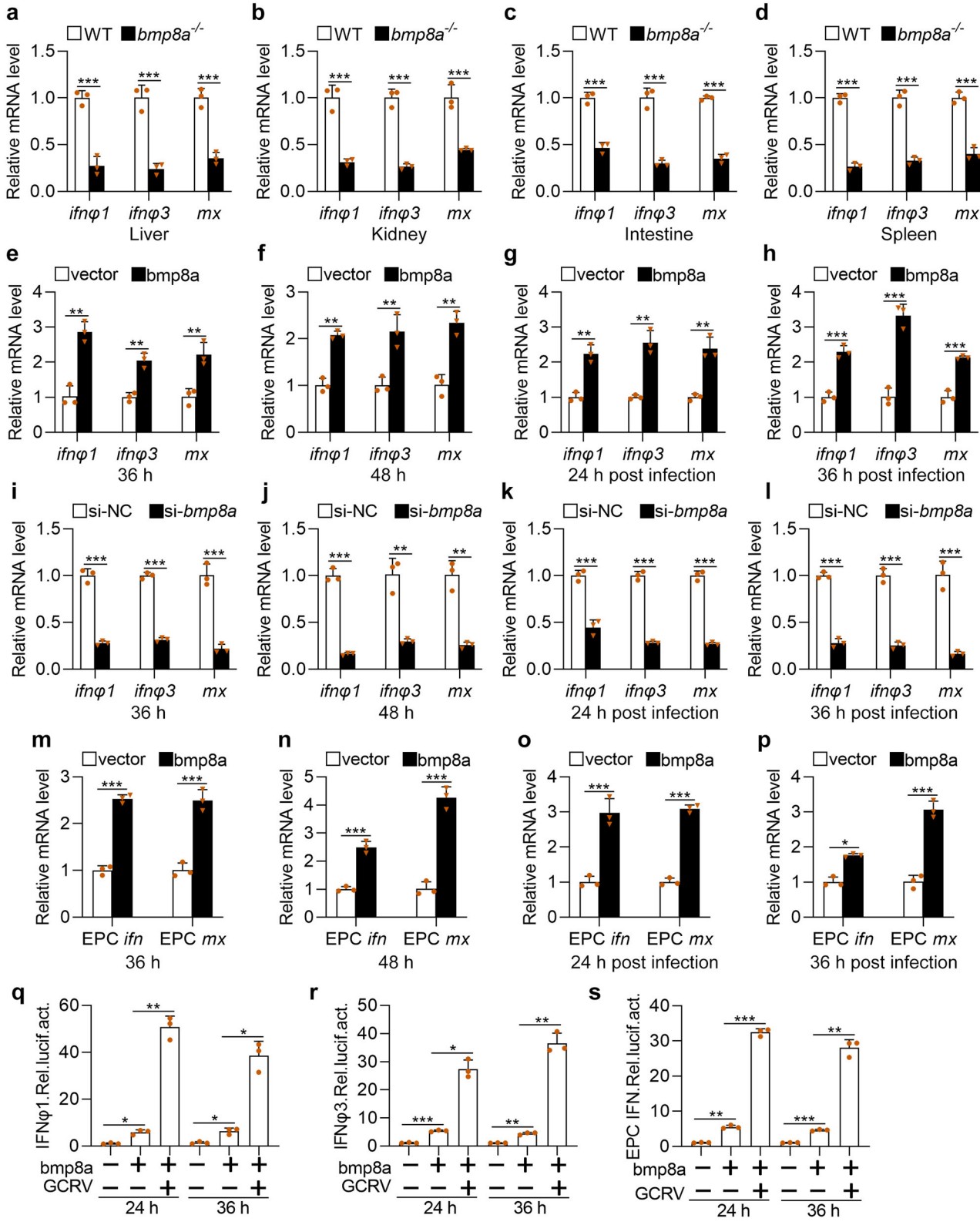

ZFL cells as well as EPC cells, infected with or without GCRV. It was found that phosphorylation levels of Tbk1 and Irf3 were significantly increased in both types of the cells with over-expressed Bmp8a (Fig. 3m–p). Moreover, *bmp8a* knockdown markedly reduced the phosphorylation levels of Tbk1 and Irf3 in ZFL cells, infected with or without GCRV (Fig. 3q, r). Further-more, in luciferase activity assays, transfection of dominant

negative mutations of *tbk1* (tbk1-K38M), *irf3* (irf3DN), or *irf7* (irf7DN) in EPC cells induced a significant loss in the ability of Bmp8a to activate the IFN promoter (Fig. 3s–u). This was further supported by the observations in vivo that the expression of *tbk1*, *irf3*, and *irf7* in the liver, kidney, intestine, and spleen was sig-nificantly downregulated in *bmp8a*[−/−] mutant than that in wild-type zebrafish (Fig. 3v–y). These data together revealed that

**Fig. 2 Bmp8a promotes antiviral innate immune responses. a–d** Expression of *ifnφ1*, *ifnφ3*, and *mx* mRNA in the liver, kidney, intestine, and spleen from WT or *bmp8a*−/− zebrafish injected i.p. with 50 µl of GCRV ($10^8$TCID$_{50}$ per ml) for 72 h. **e, f** Expression of *ifnφ1*, *ifnφ3*, and *mx* mRNA after transfected with bmp8a (2 µg) or empty vector (2 µg) in ZFL cells. The cells were collected at 36 h (**e**) or 48 h (**f**) post-transfection. **g, h** Expression of *ifnφ1*, *ifnφ3*, and *mx* mRNA after transfected with bmp8a (2 µg) or empty vector (2 µg) in ZFL cells for 24 h, followed by infection with GCRV for another 24 h (**g**) or 36 h (**h**). **i, l** Expression of *ifnφ1*, *ifnφ3*, and *mx* mRNA after *bmp8a* knockdown in ZFL cells. The cells were collected at 36 h (**i**) or 48 h (**l**) post knockdown. **k, l** Expression of *ifnφ1*, *ifnφ3*, and *mx* mRNA after knockdown *bmp8a* in ZFL cells for 24 h, followed by infection with GCRV for another 24 h (**k**) or 36 h (**l**). **m, n** Expression of EPC *ifn* and EPC *mx* mRNA after transfected with bmp8a (2 µg) or empty vector (2 µg) in EPC cells. The cells were collected at 36 h (**m**) or 48 h (**n**) post-transfection. **o, p** Expression of EPC *ifn* and EPC *mx* mRNA after transfected with bmp8a (2 µg) or empty vector (2 µg) in EPC cells for 24 h, followed by infection with GCRV for another 24 h (**o**) or 36 h (**p**). The expression of zebrafish *actb1* or EPC *actin* was used as an internal control for the qRT-PCR. **q–s** EPC cells were transfected with IFNφ1pro-luc (200 ng, **q**), IFNφ3pro-luc (200 ng, **r**) or EPC IFN pro-luc (200 ng, **s**) respectively, with or without bmp8a (200 ng), followed by infection with GCRV. At the indicated time points, cells were collected for luciferase assays. *Renilla* luciferase was used as the internal control. Data were from three independent experiments and were analyzed by Student's *t*-test (two-tailed) for comparison of two groups or one-way ANOVA followed by Games–Howell posthoc tests for comparison of multiple groups. All data were presented as mean ± SD (*$p < 0.05$, **$p < 0.01$, ***$p < 0.001$).

Bmp8a induced *ifn* and *mx* expression via promoting Tbk1–Irf3–Ifn antiviral signaling.

BMPs transmit signals through SMAD1/5/8, SMAD2/3, ERK, JNK, or p38 MAPK pathways[41–43]. Thus, the effects of p38 MAPK inhibitor SB203580, JNK inhibitor SP600125, MEK1/2 inhibitor U0126, SMAD1/5/8 inhibitor DMH1, and SMAD2/3 inhibitor TP0427736 HCl on the expression of *ifn* was tested in bmp8a-overexpressing ZFL cells and EPC cells. We have shown above that Bmp8a overexpression increased the expression of *ifn* (Fig. 2). Here, we found that only p38 MAPK inhibitor SB203580 significantly reduced the expression of *ifnφ1* and *ifnφ3* in ZFL cells and the expression of *ifn* in EPC cells (Fig. 4a–c). Furthermore, the IFN promoter-driven luciferase assays revealed that the luciferase activities were markedly reduced in EPC cells upon treatment with p38 MAPK inhibitor SB203580 (Fig. 4d–f), consistent with the observation that Bmp8a induced the expression of *ifn* through p38 MAPK pathway. Taken the previous report that the MAPK signaling increased the phosphorylation of TBK1 and IRF3 upon viral infection into consideration[44], we suggested that Bmp8a activated Tbk1-Irf3-Ifn antiviral signaling via p38 MAPK pathway.

**BMP type I receptor Alk6a interacted with Bmp8a and participated in antiviral immunity.** BMPs exert their biological effects through the sequential activation of two types of transmembrane receptors, namely BMP receptor type I and BMP receptor type II, that both possess intrinsic serine/threonine kinase activity[45,46]. To detect which receptor is involved in the regulation of the antiviral immune responses, the expression of both type I receptor genes (*alk2*, *alk3*, and *alk6a*) and type II receptor genes (*bmpr2a*, *bmpr2b*, *actr2a*, and *actr2b*) in ZFL cells transfected with poly(I:C) or infected with GCRV were measured. Among these receptor genes, *alk6a* showed the highest expression upon poly(I:C) or virus challenge (Fig. 5a, b). We also overexpressed all these receptors in ZFL and EPC cells, and examined the expression of *ifn* in these cells. The upregulation of *ifnφ1* was only found in the alk6a-overexpressing ZFL cells (Fig. 5c). Moreover, the overexpression of *alk6a* or *alk2* resulted in significantly increased expression of *ifnφ3* in ZFL cells and *ifn* in EPC cells, with overexpressed *alk6a* being more effective (Fig. 5d, e). Based on these findings, the function of Alk6a was selected for further test. Compared to wild-type ZFL cells, the expressions of antiviral protein genes *ifnφ1*, *ifnφ3*, and *mx* as well as the antiviral signaling genes *tbk1*, *irf3*, and *irf7* were all markedly upregulated in alk6a-overexpressed ZFL cells with or without GCRV infection (Fig. 5f, g). The similar results were also observed in the alk6a-overexpressed EPC cells (Fig. 5h, i). Because the phosphorylation of Gly-Ser (GS) domain of the type I receptor is required for its activation, we thus constructed dominant-negative mutant of

alk6a (alk6a-ΔGS) plasmid (Fig. 5j), and then transferred it into both ZFL and EPC cells. It revealed that the dominant-negative mutation of *alk6a* significantly reduced the expression of *mx*, *ifn* (*ifnφ1* and *ifnφ3* in ZFL cells), *tbk1*, *irf3*, and *irf7* in ZFL and EPC cells, with or without GCRV infection (Fig. 5k–n). The data suggested that Alk6a was apparently involved in the Tbk1–Irf3/7-Ifn antiviral signaling.

Considering we have revealed that Bmp8a activated Tbk1–Irf3–Ifn antiviral signaling via p38 MAPK pathway, we then examined whether the the levels of phosphorylated p38 MAPK protein were regulated by Alk6a. Clearly, compared to wild-type ZFL or EPC cells, phosphorylation levels of p38 MAPK were significantly increased in alk6a-overexpressed both types of the cells with or without GCRV infection (Fig. 6a–d). Moreover, dominant-negative mutation of *alk6a* significantly reduced the phosphorylation levels of p38 MAPK in ZFL or EPC cells, with or without GCRV infection (Fig. 6e–h). Thus, Alk6a was shown to be involved in the phosphorylation of p38 MAPK. Next, we investigated whether the antiviral immune responses of Bmp8a were mediated by Alk6a. In the IFN promoter-driven luciferase assays, it was found that the ability of Bmp8a to activate the IFN promoter was markedly blocked after the dominant-negative mutation of *alk6a* in EPC cells (Fig. 6i–k). Importantly, co-immunoprecipitation (Co-IP) experiments confirmed that Bmp8a interacted with Alk6a (Fig. 6l). Collectively, our data demonstrated that BMP type I receptor Alk6a interacted with Bmp8a, and participated in the antiviral immunity. Alk6a is the first BMP receptor identified thus far which is being directly required for antiviral immune responses.

**Virus induced expression of *bmp8a*.** Next, we measured the expression of *bmp8a* in ZFL cells upon virus infection. GCRV significantly induced the expression of *bmp8a* at 12 h of post-stimulation, and reached a peak at 48 h (Fig. 7a). We also detected the expression of *bmp8a* upon infection with GCRV or poly(I:C) in vivo. The expression of *bmp8a* in the liver, kidney, intestine, and spleen of zebrafish infected with GCRV or poly(I:C) was all significantly elevated (Fig. 7b–i). This indicated that the expression of *bmp8a* was inducible by infection with virus or its mimic poly(I:C).

**Binding of Stat1 to GAS motifs activated *bmp8a* expression.** To explore how virus induces expression of *bmp8a*, we searched for the transcription factor binding sites in *bmp8a* promoter region at the web http://jaspar.genereg.net/. Two gamma-activated sites (GAS), 5′-ATTCCGGGAAA-3′ (P1) and 5′-TTTACTAGAAC-3′ (P2), in *bmp8a* promoter region were identified (Supplementary Fig. 3). In mammals, the transcription factor STAT1 homodimer

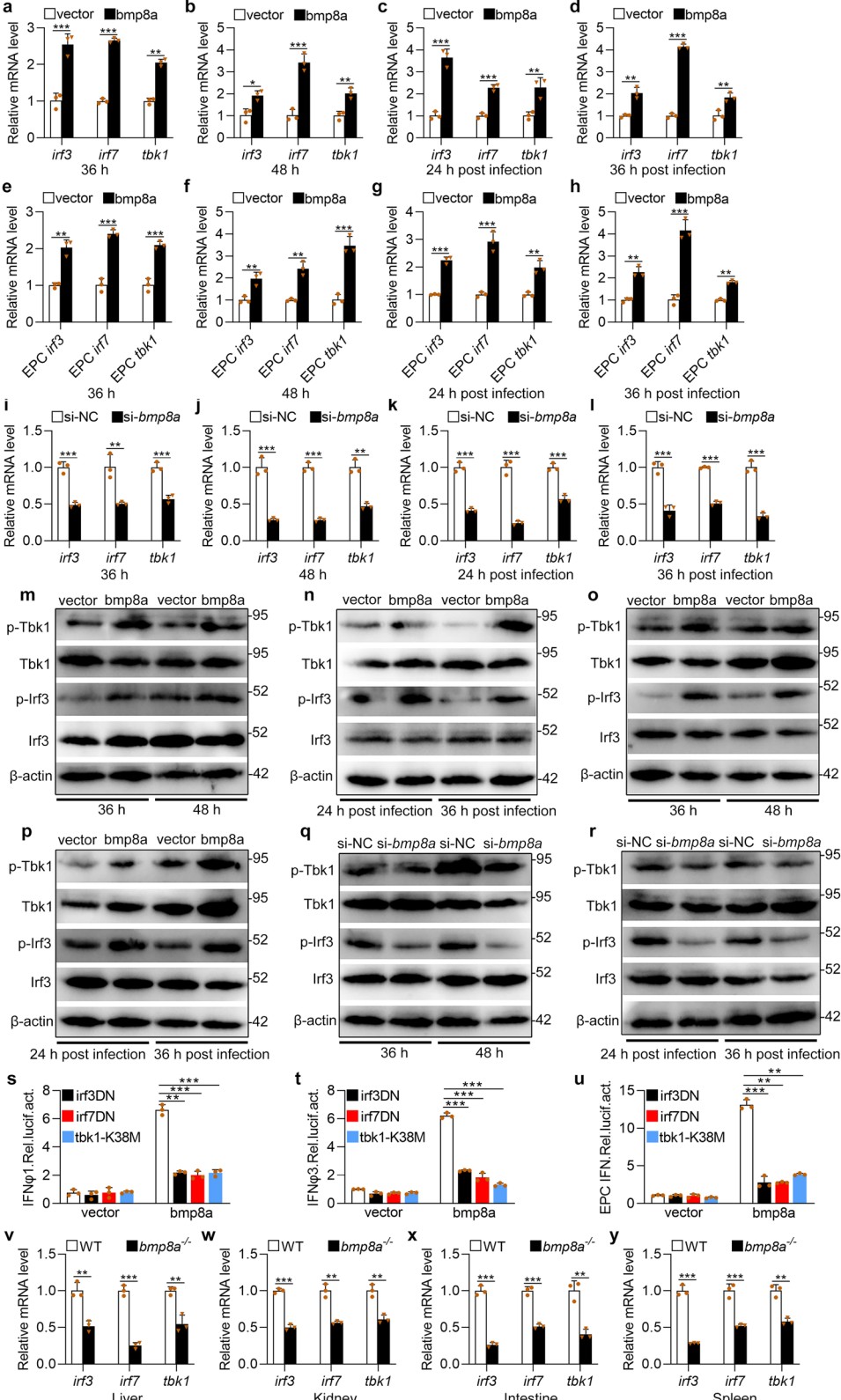

was known to bind to the GAS motif in the promoter region of IFN-γ triggered downstream genes[11,16]. We thus wonder if Stat1 can bind to the *bmp8a* promoter region to activate the expression of *bmp8a*. In zebrafish, there exist two Stat1 factors, Stat1a (GenBank accession number NM_131480.1) and Stat1b (GenBank accession number NM_200091.2). Thus we constructed dominant-negative mutant plasmids of stat1a-ΔC and stat1b-ΔC,

that were transferred into ZFL cells, respectively. It was found that the expression of *bmp8a* was significantly blocked in both cases (Fig. 8a, b). By contrast, the overexpression of either *stat1a* or *stat1b* markedly promoted *bmp8a* promoter-driven luciferase activities (Fig. 8c). In addition, when one of the GAS motifs in *bmp8a* promoter region was deleted, *bmp8a* promoter-driven luciferase activities were significantly reduced; when both of the

**Fig. 3 Bmp8a increases Tbk1–Irf3–Ifn antiviral signaling. a, b, e, f** Expression of *irf3, irf7,* and *tbk1* mRNA after transfected with 2 μg bmp8a or empty vector in ZFL (**a, b**) or EPC (**e, f**) cells. The cells were collected at 36 h (**a, e**) or 48 h (**b, f**) post-transfection. **c, d, g, h** Expression of *irf3, irf7,* and *tbk1* mRNA after transfected with 2 μg bmp8a or empty vector in ZFL (**c, d**) or EPC (**g, h**) cells for 24 h, followed by infection with GCRV for another 24 h (**c, g**) or 36 h (**d, h**). **i–l** Expression of *irf3, irf7,* and *tbk1* mRNA after *bmp8a* knockdown in ZFL cells. The cells were collected at 36 h (**i**) and 48 h (**j**) post-knockdown or at 24 h (**k**) and 36 h (**l**) post-infected with GCRV. **m, o** Immunoblot analysis of phosphorylated (p-) Tbk1 and Irf3 after transfected with 2 μg bmp8a or empty vector in ZFL (**m**) or EPC (**o**) cells. The cells were collected at 36 or 48 h post-transfection for Immunoblot analysis. **n, p** Immunoblot analysis of phosphorylated (p-) Tbk1 and Irf3 after transfected with 2 μg bmp8a or empty vector in ZFL (**n**) or EPC (**p**) cells for 24 h, followed by infection with GCRV for another 24 or 36 h. **q, r** Immunoblot analysis of phosphorylated (p-) TBK1 and IRF3 after *bmp8a* knockdown in ZFL cells. The cells were collected at 36 and 48 h post-knockdown or at 24 and 36 h post-infected with GCRV. **s–u** EPC cells were cotransfected with IFN-φ1pro-luc (200 ng, **s**), IFN-φ3pro-luc (200 ng, **t**) or EPC IFNpro-luc (200 ng, **u**), and bmp8a (100 ng) together with each of the dominant negative plasmids including tbk1-K38M (100 ng), irf3DN (100 ng) and irf7DN (100 ng). At 48 h post-transfection, the cells were collected for luciferase assays. *Renilla* luciferase was used as the internal control. **v–y** Expression of *irf3, irf7,* and *tbk1* mRNA in the liver, kidney, intestine, and spleen from WT or *bmp8a*$^{-/-}$ zebrafish injected i.p. with 50 μl of GCRV ($10^8$TCID$_{50}$ per ml). The expression of zebrafish *actb1* or EPC *actin* was used as an internal control for the qRT-PCR. Data were from three independent experiments and were analyzed by Student's *t*-test (two-tailed) for comparison of two groups or one-way ANOVA followed by Games–Howell post hoc tests for comparison of multiple groups. All data were presented as mean ± SD (**$p < 0.01$, ***$p < 0.001$).

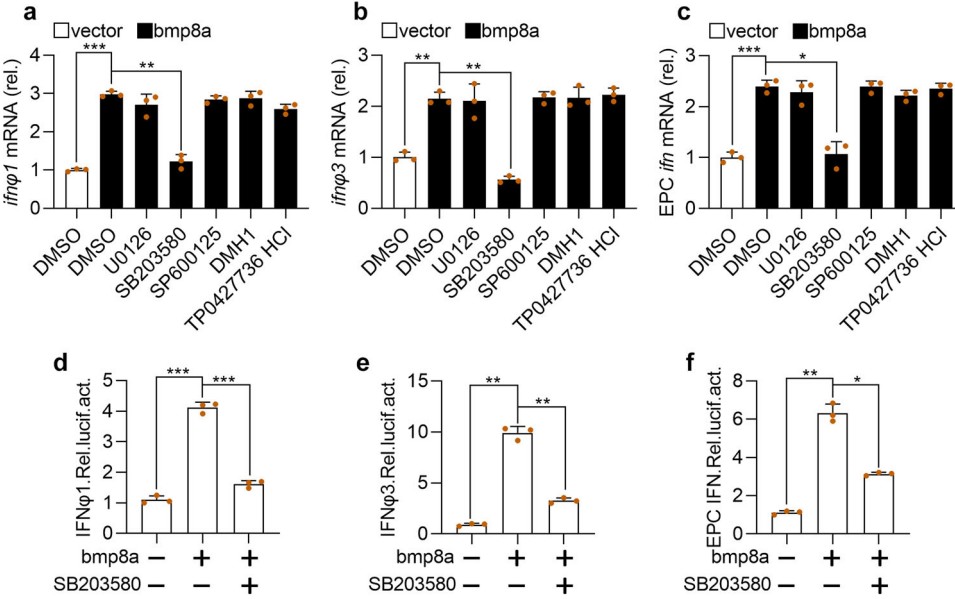

**Fig. 4 Bmp8a promotes the IFN expression via p38 MAPK pathway. a, b** Expression of *ifnφ1* (**a**) and *ifnφ3* (**b**) mRNA after transfected with bmp8a (2 μg) in ZFL cells for 24 h, followed by treatment with SB203580, SP600125, U0126, DMH1, and TP0427736 HCl for another 24 h. **c** Similar as (**a, b**) but in EPC cells. The expression of zebrafish *actb1* or EPC *actin* was used as an internal control for the qRT-PCR. **d–f** EPC cells were cotransfected with IFNφ1pro-luc (200 ng, **d**), IFNφ3pro-luc (200 ng, **e**) or EPC IFNpro-luc (200 ng, **f**), pRL-TK (20 ng) together with bmp8a (200 ng) or empty vector (200 ng), respectively. At 24 h of post-transfection, cells were treated with or without SB203580 for another 24 h and then harvested for detection of luciferase activity. *Renilla* luciferase was used as the internal control. Data were from three independent experiments and were analyzed by Student's *t*-test (two-tailed) for comparison of two groups or one-way ANOVA followed by Games–Howell post hoc tests for comparison of multiple groups. All data were presented as mean ± SD (*$p < 0.05$, **$p < 0.01$, and ***$p < 0.001$).

GAS motifs were deleted, *bmp8a* promoter-driven luciferase activities were reduced even more significantly (Fig. 8d, e). These suggested that both Stat1a and Stat1b may interact with the GAS motifs.

To verity the binding of Stat1a/Stat1b to the GAS motifs in *bmp8a* promoter region, the recombinant proteins of both Stat1a and Stat1b were expressed and purified (Fig. 8f, g), and used for electrophoretic mobility shift assay (EMSA). As shown in Fig. 8h–k, a remarkable band of protein–DNA complex was observed (lane 2). The protein–DNA complex band was obviously weaker in the presence of an excess unlabeled competitor probe, compared to that of biotin labeled probe group (lane 3), but the intensity of the band showed little change in the presence of GAS mutant probes (lane 4). To detect the specific binding, a supershift experiment with the antibody against His tag was performed. It revealed that His tag antibody caused a specific supershift of slower migrating protein–DNA–antibody complex (Fig. 8h–k,

lane 5). There was no protein–DNA complex observed in the negative control group (Fig. 8h–k, lane 1). These demonstrated that both Stat1a and Stat1b could directly bind to the GAS motifs in *bmp8a* promoter region, suggesting that *bmp8a* expression was subjected to the regulation by binding of Stat1a/Stat1b to the GAS motifs.

## Discussion

BMPs, canonical multifunctional growth factors, are suggested to play roles in regulation of immune responses, but it remains controversial over their functions in immunity. We show here for the first time that Bmp8a is a previously unrecognized factor involved in the regulation of antiviral immune responses in zebrafish. Evidences supporting this nature of Bmp8a include: Bmp8a inhibits RNA viruses replication in vitro and in vivo; Bmp8a promotes the expression of antiviral protein genes

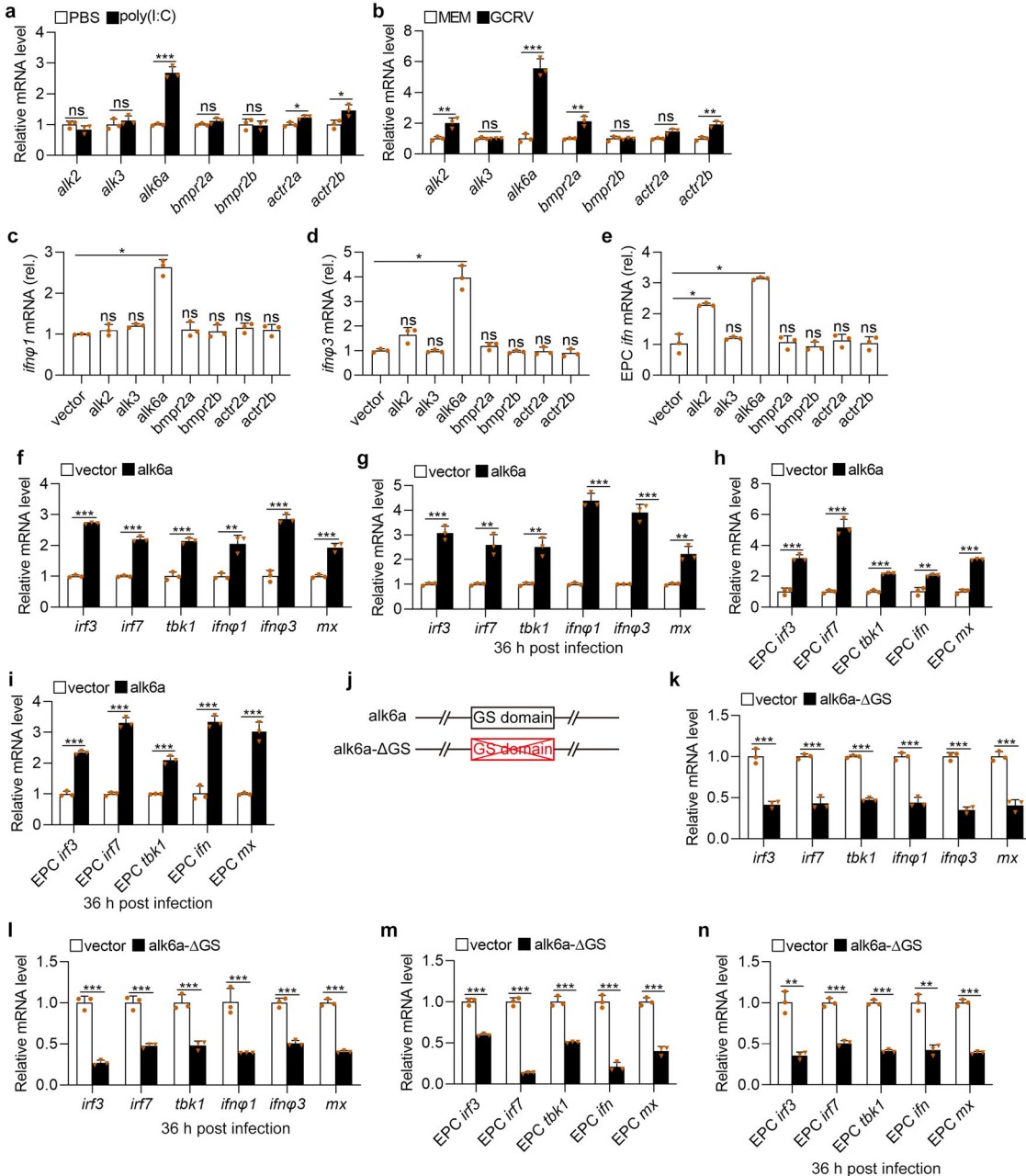

**Fig. 5 Alk6a is involved in the antiviral innate immune responses. a, b** Expression of *alk2*, *alk3*, *alk6a*, *bmpr2a*, *bmpr2b*, *actr2a*, and *actr2b* mRNA in ZFL cells stimulated with poly(I:C) (2 µg/ml, **a**) or GCRV ($5 \times 10^4$TCID$_{50}$ per ml, **b**) for 48 h. **c–e** Expression of *ifnφ1* (**c**) and *ifnφ3* (**d**) mRNA in ZFL cells or EPC *ifn* (**e**) in EPC cells which were transfected with 2 µg of alk2, alk3, alk6a, bmpr2a, bmpr2b, actr2a, actr2b or empty vector for 48 h. **f, h** Expression of *irf3*, *irf7*, *tbk1*, *ifn* (or *ifnφ1* and *ifnφ3*), and *mx* mRNA after transfected with 2 µg of pcDNA3.1-alk6a or empty vector in ZFL (**f**) or EPC (**h**) cells for 48 h. **g, i** Expression of *irf3*, *irf7*, *tbk1*, *ifn* (or *ifnφ1* and *ifnφ3*), and *mx* mRNA after transfected with 2 µg of of pcDNA3.1-alk6a or empty vector in ZFL (**g**) or EPC (**i**) cells for 24 h, followed by infection with GCRV for another 36 h. **j** Schematic drawing of the alk6a-ΔGS mutation that the GS domain of Alk6a was deleted. **k, m** Expression of *irf3*, *irf7*, *tbk1*, *ifn* (or *ifnφ1* and *ifnφ3*), and *mx* mRNA after transfected with 2 µg pcDNA3.1-alk6a-ΔGS or empty vector in ZFL (**k**) or EPC (**m**) cells for 48 h. **l, n** Expression of *irf3*, *irf7*, *tbk1*, *ifn* (or *ifnφ1* and *ifnφ3*), and *mx* mRNA after transfected with 2 µg pcDNA3.1-alk6a-ΔGS or empty vector in ZFL (**l**) or EPC (**n**) cells for 24 h, followed by infection with GCRV for another 36 h. The expression of zebrafish *actb1* or EPC *actin* was used as an internal control for the qRT-PCR. Data were from three independent experiments and were analyzed by Student's *t*-test (two-tailed) for comparison of two groups or one-way ANOVA followed by Games–Howell posthoc tests for comparison of multiple groups. All data were presented as mean ± SD (\**p* < 0.05, \*\**p* < 0.01, and \*\*\* *p* < 0.001, ns means no significant difference).

including type I IFNs and *mx*; the expression of *bmp8a* is induced by CGRV or poly(I:C), and Bmp8a activates TBK1–IRF3–IFN signaling. Bmp8a is a member of the TGF-β superfamily, which includes more than 30 genes encoding TGFβ, BMPs and activins. TGFβ has been shown to have both pro-inflammatory and anti-inflammatory effects depending on the context it is acting in the immune system[47]. Like TGFβ, BMP members have also been shown to play different immunoregulatory roles[48]. BMP2, 4 and 7 deficiency or knockdown results in partial loss of the thymic capsule, reduced size of thymus and increased *Helicobacter pylori*-induced inflammation, whereas BMP6 blocks HCV replication and inhibits macrophage growth by inducing cell cycle arrest

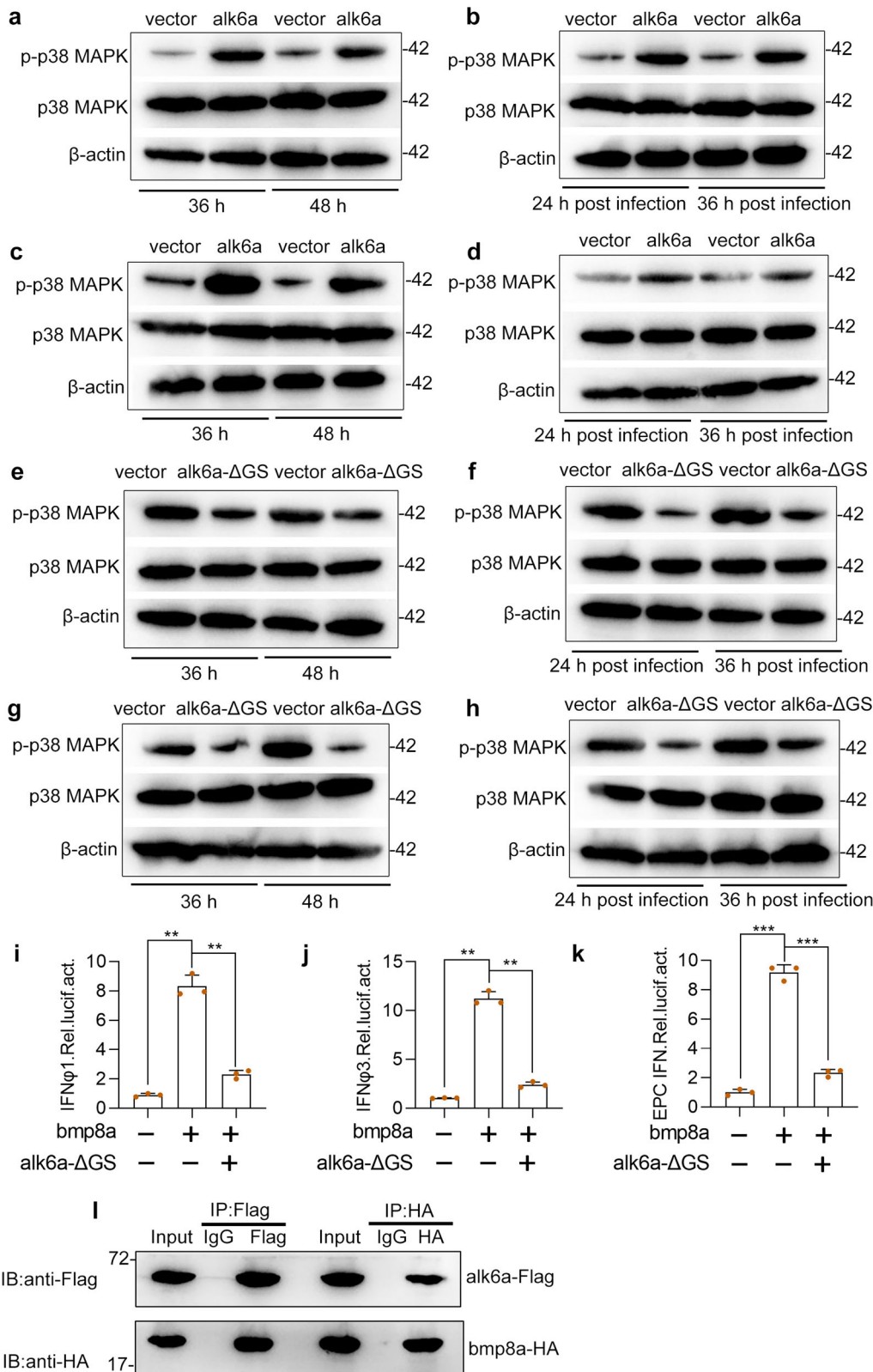

in vitro[27,29,49-52]. The finding that Bmp8a is a positive regulator in antiviral immune responses provides a angle for our understanding of the functions of TGF-β superfamily members including BMPs.

On one hand, it is generally regarded that the recognition of virus by PRRs elicits the activation of IRF3 and NF-κB, and induces the production of IFNα/β, that then activate JAK-STAT pathway, eventually leading to the production of ISGs such as cytokines and antiviral proteins. On the other hand, BMPs are known to transmit signals through both SMAD-dependent pathways (SMAD1/5/8 and SMAD2/3 pathways) and SMAD-independent pathways (ERK, JNK, and p38 MAPK pathways). Here we clearly demonstrate that Bmp8a promotes the expression of endogenous type I IFNs independent of SMAD signaling

**Fig. 6 Bmp8a activates the IFN expression through Alk6a. a, c** Immunoblot analysis of phosphorylated (p-) p38 MAPK after transfected with 2 µg alk6a or empty vector in ZFL (**a**) or EPC (**c**) cells. The cells were collected at 36 or 48 h of post-transfection for Immunoblot analysis. **b, d** Immunoblot analysis of phosphorylated (p-) p38 MAPK after transfected with 2 µg alk6a or empty vector in ZFL (**b**) or EPC (**d**) cells for 24 h, followed by infection with GCRV (5 × $10^4$ TCID$_{50}$ per ml) for another 24 h or 36 h. **e, g** Immunoblot analysis of phosphorylated (p-) p38 MAPK after transfected with 2 µg alk6a-ΔGS or empty vector in ZFL (**e**) or EPC (**g**) cells. The cells were collected at 36 or 48 h of post-transfection for Immunoblot analysis. **f, h** Immunoblot analysis of phosphorylated (p-) p38 MAPK after transfected with 2 µg alk6a or empty vector in ZFL (**f**) or EPC (**h**) cells for 24 h, followed by infection with GCRV (5 × $10^4$ TCID$_{50}$ per ml) for another 24 or 36 h. **i–k** EPC cells were cotransfected with IFN-φ1pro-luc (200 ng, **i**), IFN-φ3pro-luc (200 ng, **j**) or EPC IFNpro-luc (200 ng, **k**), and bmp8a (100 ng) together with or without the dominant negative plasmids alk6a-ΔGS (100 ng). At 48 h of post-transfection, the cells were collected for luciferase assays. *Renilla* luciferase was used as the internal control. **l** Co-immunoprecipitation and immunoblot analysis of EPC cells cotransfected with alk6a-Flag (1 µg) and bmp8a-HA (1 µg). Data were from three independent experiments and were analyzed by Student's *t*-test (two-tailed) for comparison of two groups or one-way ANOVA followed by Games–Howell posthoc tests for comparison of multiple groups. All data were presented as mean ± SD (**$p < 0.01$, ***$p < 0.001$).

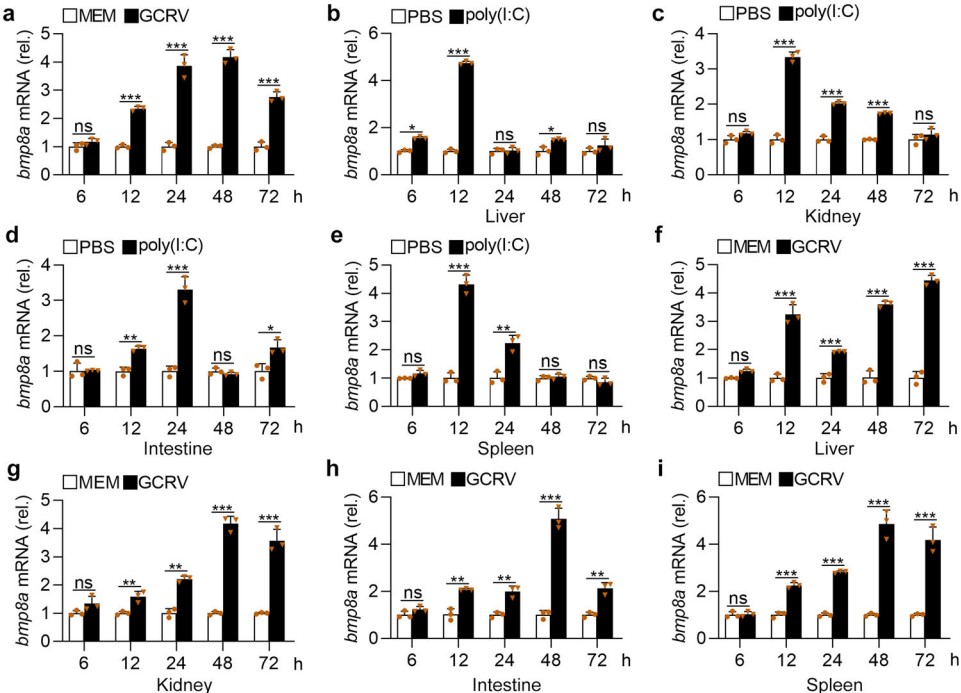

**Fig. 7 GCRV or poly(I:C) increases the expression of *bmp8a*. a** Expression of *bmp8a* mRNA in ZFL cells after infected with GCRV (5 × $10^4$ TCID$_{50}$ per ml). **b–e** Expression of *bmp8a* mRNA in the liver (**b**), spleen (**c**), intestine (**d**), and kidney (**e**) from zebrafish injected i.p. with poly(I:C) (10 µg/fish). **f–i** Expression of *bmp8a* mRNA in the liver (**f**), spleen (**g**), intestine (**h**), and kidney (**i**) from zebrafish injected i.p. with 50 µl of GCRV ($10^8$ TCID$_{50}$ per ml). Zebrafish injected i.p. with PBS or MEM were used as the control. The expression of *actb1* served as a control for the qRT-PCR. Data were from three independent experiments and were analyzed by Student's *t*-test (two-tailed). All data were presented as mean ± SD (*$p < 0.05$, **$p < 0.01$, and ***$p < 0.001$, ns means no significant difference).

pathways. Moreover, it is p38 MAPK pathway, rather than ERK and JNK pathways, that is involved in the antiviral responses induced by Bmp8a. Furthermore, we show that Alk6a participates in Tbk1–Irf3/7–Ifn antiviral signaling via interaction with Bmp8a. In addition, Alk6a was found to be required to induce the phosphorylation of p38 MAPK, which is consistent with the findings in mouse that Alk6 knockdown suppresses p38 MAPK phosphorylation[53]. Previous studies have shown that MAPK signaling pathway is crucial in the phosphorylation of TBK1 and IRF3 upon viral infection[44]. Thus, we probably discover a pathway of Bmp8a signaling in antiviral immune responses that Bmp8a acts as a positive regulator through the promotion of phosphorylation of Tbk1 via p38 MAPK pathway, i.e., Bmp8a binds to Alk6a, which induces p38 MAPK phosphorylation, and in turn enhances Tbk1 and Irf3 phosphorylation, eventually resulting in increased synthesis of type I IFN (Fig. 9). However, more recently, BMP6 has been suggested to suppress HCV replication independently of IFN, although it can also enhance

the transcriptional and antiviral response to IFN[29]. Therefore, we can not rule out the possibility that Bmp8a has antiviral activity independent of p38 MAPK–Tbk1–Irf3/7–IFN pathway.

Our study also shows that the challenge with GCRV and poly (I:C) both significantly promotes the expression of zebrafish *bmp8a*. Searching for the binding sites of transcript factors in *bmp8a* promoter region reveals that two IFN-γ activation sites (GAS) that may interact with STAT1 are identified. Functionally, we demonstrate that Stat1a and Stat1b, orthologues of human STAT1, can directly bind the GAS sites in *bmp8a* promoter, resulting in activation of *bmp8a* expression. It has been reported that in black carp, Stat1a and Stat1b can both form homodimer and heterodimer in vivo like their mammalian counterpart[54]. It is thus highly likely that in zebrafish, Stat1a and Stat1b can form homodimer or heterodimer, which then binds to the GAS sites in *bmp8a* promoter region. In zebrafish, IFN-γ has been found to be able to induce the expression of both *stat1a* and *stat1b*, and the GAS motif is required for Ifn-γ activation[55,56]. It is widely known

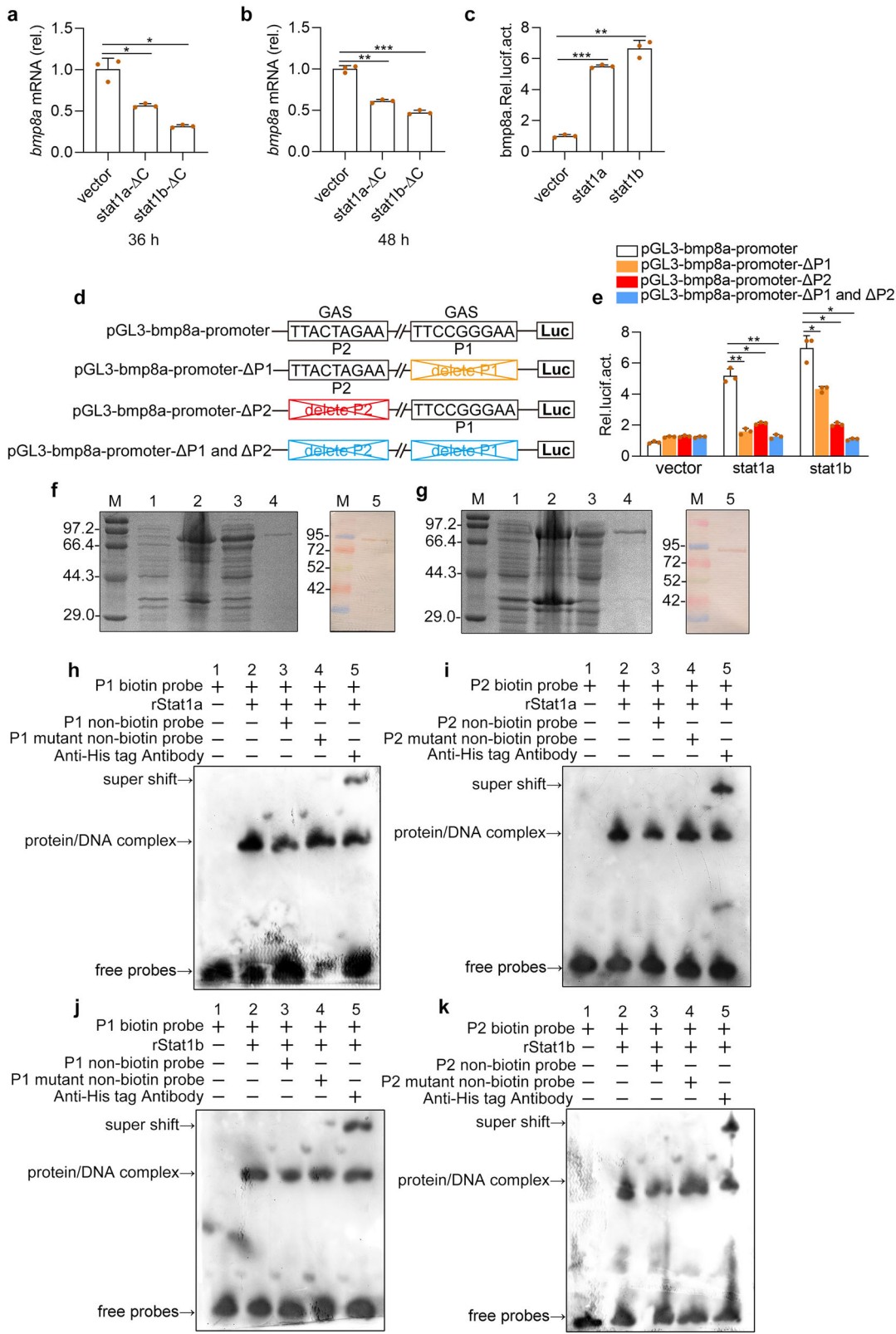

COMMUNICATIONS BIOLOGY | (2021)4:318 | https://doi.org/10.1038/s42003-021-01811-0 | www.nature.com/commsbio

that the JAK-STAT pathway is one of the most important candidate pathways through which IFN-γ works. Activation of JAK-STAT signaling leads to transcription of various downstream ISGs for antiviral activity[57]. We thus think that Bmp8a is another important antiviral factor which can be activated through Ifn-γ-Jak-Stat1 pathway (Fig. 9).

In summary, we demonstrate for the first time that Bmp8a is a positive regulator in antiviral immune responses, which functions through promotion of phosphorylation of Tbk1 via p38 MAPK pathway. Upon virus infection, the expression of *bmp8a* is activated by Stat1a/Stat1b directly. Bmp8a knockdown leads to significantly decreased phosphorylation of Tbk1 and Irf3. As a

**Fig. 8 Stat1 binds the GAS sites and activates the *bmp8a* transcriptions upon virus stimulation. a, b** Expression of *bmp8a* mRNA in ZFL cells after transfected with stat1a-ΔC or stat1b-ΔC for 36 h (**a**) or 48 h (**b**). The expression of *actb1* served as a control for the qRT-PCR. **c** Dual luciferase report assay was used to analyze the transcription abilities of Stat1a and Stat1b in activation of *bmp8a* in EPC cells. pGl3-bmp8a (200 ng) was transfected into EPC cells together with stat1a (200 ng), stat1b (200 ng), or empty vector (200 ng). After 48 h, the transfected cells were collected for luciferase assays. **d** Schematic drawing of wild-type and GAS motif mutation Luc-report plasmids. **e** The 200 ng of pGL3-bmp8a-promoter, pGL3-bmp8a-promoter-ΔP1, pGL3-bmp8a-promoter-ΔP2 or pGL3-bmp8a-promoter-ΔP1 and ΔP2 was transfected into EPC cells along with stat1a (200 ng), stat1b (200 ng) or empty vector (200 ng), respectively. After 48 h, the transfected cells were collected for luciferase assays. *Renilla* luciferase was used as the internal control. **f, g** SDS-PAGE and Western-blotting analysis of rStat1a (**f**) and rStat1b (**g**). Lane M: protein molecular standard; Lane 1: negative control for IPTG induced *E. coli* (without rStat1a or rStat1b); Lane 2: induced rStat1a or rStat1b (the whole cell lysate); Lane 3: induced rStat1a or rStat1b (the supernatant); Lane 4: purified rStat1a or rStat1b; Lane 5: western blot analysis of the sample in Lane 4. **h–k** EMSA was performed to validate the interaction of rStat1a or rStat1b with the GAS motif (P1 or P2) in the bmp8a promoter region. Lane 1: negative group; Lane 2: positive group; Lane 3: an excess unlabeled competitor probe; Lane 4: an excess unlabeled competitor probe containing a mutated runt binding site; Lane 5: Super-shift assays were performed by adding antibody against His tag. Data were from three independent experiments and were analyzed by Student's *t*-test (two-tailed) for comparison of two groups or one-way ANOVA followed by Games–Howell posthoc tests for comparison of multiple groups. All data were presented as mean ± SD (*$p < 0.05$, **$p < 0.01$, and ***$p < 0.001$).

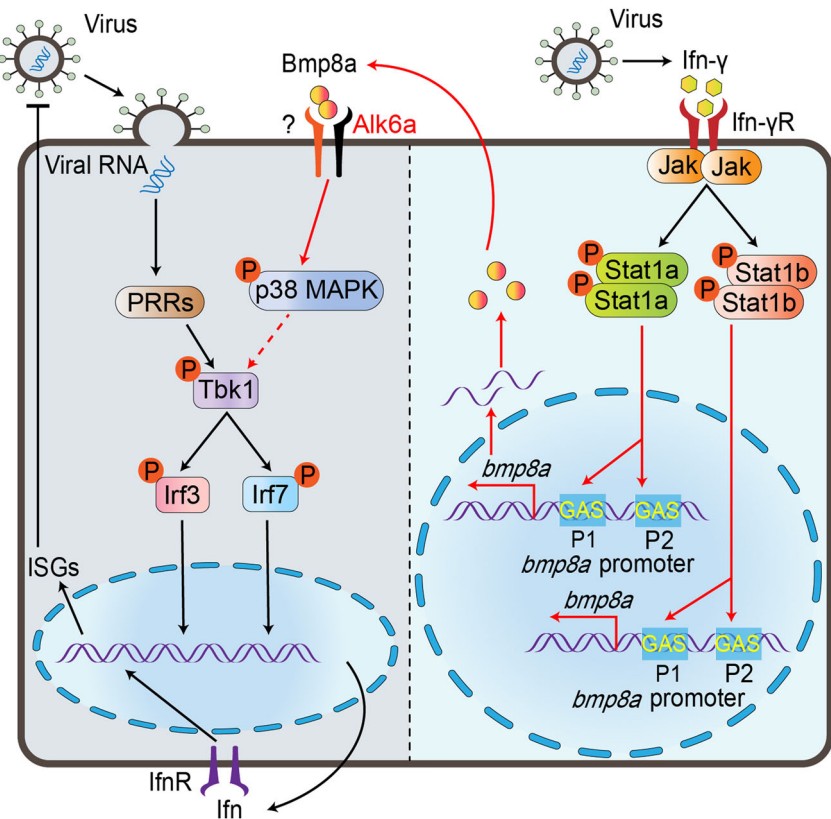

**Fig. 9 Schematic illustration of Bmp8a serving as positive regulator in the antiviral immune.** Upon virus infection, the transcriptions of *bmp8a* are activated through the Jak-Stat1 pathway. The Bmp8a binds to BMP type I receptor Alk6a, promoting phosphorylation of Tbk1 and Irf3 to induce the expression of *Ifn* through p38 MAPK pathway.

consequence, *bmp8a*-deficient zebrafish produces lowered type I IFNs in response to RNA virus infection, exhibits considerably reduced immune responses, and increases higher viral load and mortality in vivo. Our study is important to understand the regulatory mechanism of BMP involving in antiviral immune response and provides a target for controlling viral infection.

## Methods

**Cells and viruses**. The epithelioma papulosum cyprini cells (EPC), zebrafish liver cells (ZFL) were acquired from the China Zebrafish Resource Center (CZRC) and cultured according to the instructions from the CZRC. EPC and ZFL cells were grown in a 28 °C incubator supplied with 5% CO₂. The flounder gill cells (FG) were grown in a 22 °C incubator supplied. The EPC or FG cells were cultured in MEM media supplemented with 10% fetal bovine serum (FBS) (Gibco), 100 U/ml penicillin and 100 µg/ml streptomycin. The ZFL cells were maintained in DMEM/

F-12 media supplemented with 10% FBS, 100 U/ml penicillin, and 100 µg/ml streptomycin. Grass carp reovirus (GCRV), a dsRNA virus, and spring viremia of carp virus (SVCV), a negative ssRNA virus, were kindly provided by Yibing Zhang (Institute of Hydrobiology, Chinese Academy of Sciences). The virus titer was determined by a 50% tissue culture-infective dose (TCID₅₀) assay on EPC cells according to the method of Reed and Muench[58]. In brief, EPC cells were seeded in a 96-well cell culture plates. By gradient dilution, the virus stocks were diluted from $10^{-1}$ to $10^{-8}$. Then, the cells were inoculated with virus. The ten parallel samples are set for each gradient. Five days later, we checked all wells if they contain positive cells under microscope. Finally, Reed and Muench calculation formula was then performed to determine the TCID₅₀. The turbot skin verruca disease virus (TSVDV) was isolated from the focal site of turbot skin in our lab. All experiments related to the virus were conducted in the biosafety cabinet (BL-2 level).

**Zebrafish**. The animals used in the experiment followed the ethical guidelines established by the Institutional Animal Care and Use Committee of the Ocean University of China (permit number, SD2007695). The zebrafish *bmp8a* knockout

mutant lines ($bmp8a^{-/-}$) were established from the zebrafish AB line using TALENs technology. The TALENs for zebrafish $bmp8a$ were assembled using the Unit Assembly (UA) method. All the UA method starting vectors were the gifts from laboratories of Bo Zhang at Peking University and of Shuo Lin at University of California Los Angeles. TALENs mRNAs were transcribed using the mMES-SAGE mMACHINE SP6 kit (Invitrogen, #AM1340) and purified using the RNeasy Mini kit (QIAGEN, #74106). To generate zebrafish mutant lines, TALEN mRNAs (50–200 pg) were microinjected into 1-cell stage zebrafish embryos. Two days after injection, genomic DNA was isolated from 8 to 10 pooled larvae. The target genomic regions were amplified by nested PCR and subcloned into the pGEM-T vector. Single colonies were genotyped by sequencing. To obtain germline mutations, the TALEN-injected embryos were raised to adulthood and outcrossed with wild-type (WT) fish. The F1 progeny were genotyped by sequencing. To obtain homozygous mutants, heterozygous mutant of the same mutation were obtained and self-crossed. The primers used in this study were listed in Supplemental Table 1.

**Plasmid construction.** The open reading frames (ORF) of zebrafish bmp8a, stat1a, stat1b, alk2, alk3, alk6a, actr2a, actr2b, bmpr2a, and bmpr2b were cloned into the pcDNA3.1 expression vector for the eukaryotic expression. The ORF of zebrafish stat1a and stat1b were cloned into the pET-28a expression vector for the expression of recombinant proteins. The ORF of zebrafish bmp8a was cloned into pCMV-C-HA expression vector for the eukaryotic expression. The ORF of zebrafish alk6a was cloned into pCMV-C-Flag expression vector for the eukaryotic expression. The promoter regions of bmp8a, bmp8a-ΔP1 (delete the GAS motif of 5′-attccgggaaa-3′ in the bmp8a promoter), bmp8a-ΔP2 (delete another GAS motif of 5′-tttactagaac-3′ in the bmp8a promoter), bmp8a-ΔP1and ΔP2 (delete both the two GAS motifs), IFNφ1, IFNφ3, and EPC IFN were cloned into the pGL3-basic vector. Dominant negative mutant plasmids including tbk1-K38M (the 38th amino acid(aa) was mutated from K to M), stat1a-ΔC(aa 1–654), stat1b-ΔC(aa 1–675), irf3DN(aa 117–427), and irf7DN(aa 117–423) were cloned into the pcDNA3.1 expression vector for the eukaryotic expression[59–61]. Dominant negative mutant plasmids alk6a-ΔGS (delete the sequence of 5′-agctccggttctggctcagga-3′ encoding the GS domain) was cloned into the pcDNA3.1 expression vector. The deletions and mutations were created using Mut Express II Fast Mutagenesis Kit V2 (Vazyme, #C214-01). The primers used in this study were listed in Supplemental Table 2.

**Viral infection in vitro.** EPC, FG, or ZFL cells were plated 24 h before infection. Cells were infected with GCRV ($5 \times 10^4$ TCID$_{50}$ per ml) or TSVDV (crude virus extracts filtered by a 0.45 μm microporous membrane) for 1 h in medium without FBS. After infection, cells were washed with PBS and then medium was added with FBS. The cells and cell-free supernatants were harvested at the indicated times.

**Viral infection in vivo.** For in vivo viral infection, the adult WT and $bmp8a^{-/-}$ zebrafish were infected via intraperitoneal injection using 50 μl of SVCV ($10^8$ TCID$_{50}$ per ml), GCRV ($10^8$ TCID$_{50}$ per ml), or TSVDV (crude virus extracts filtered by a 0.45 μm microporous membrane) per fish. Total RNA was extracted and viral RNA expression in the liver, kidney, intestine, and spleen were examined by qRT-PCR. For the survival experiments, zebrafish were monitored for survival after SVCV, GCRV, or TSVDV infection.

**Crystal violet staining.** EPC or ZFL cells were transfected with bmp8a or empty vector plasmid. At 24 h of post-transfection, cells were treated with GCRV ($5 \times 10^4$ TCID$_{50}$ per ml). At 72 h after virus challenge, cells were fixed with 4% formaldehyde for 1 h and stained with 0.5% crystal violet for 2 h. After washing with tap water, visible plaques were photographed.

**Luciferase assays.** EPC cells were seeded in 24-well plates overnight and cotransfected with various plasmids using FuGENE HD Transfection Reagent (Promega, #E2311) following the manufacturer's instruction. If necessary, the cells were infected with GCRV ($5 \times 10^4$ TCID$_{50}$ per ml). At the indicated time points, luciferase activities were measured with the Luc-Pair Duo-Luciferase Assay Kits 2.0 (iGene Biotechnology, #LF002). Data were normalized by calculating the ratio between firefly luciferase activity and *Renilla* luciferase activity. The primers used for cloning the promoter are shown in Supplemental Table 2.

**RNA interference.** Small interfering RNA (siRNA) of *bmp8a* and si-NC (negative control) were purchased from RiboBio. The oligonucleotide sequences (siRNA) specific targeting *bmp8a* mRNA was as follows: 5′-GTTCTTCAGAGCTAGTCAA-3′. Transfection was performed with Lipofectamine RNAiMAX (Invitrogen, #13778075) according to the manufacturer's protocols. Total RNA was extracted for qRT-PCR analysis.

**Drug treatment and analysis.** SMAD1/5/8 inhibitor DMH1(Selleck, #S7146), SMAD2/3 inhibitor TP0427736 HCl (Selleck, #S8700), p38 MAPK inhibitor SB203580 (Beyotime, #S1863), JNK inhibitor SP600125 (Beyotime, #S1876), and MEK1/2 inhibitor U0126 (Beyotime, #S1901) were dissolved in dimethyl ulfoxide (DMSO). ZFL or EPC cells were treated with each inhibitor, which were diluted

with culture medium at concentrations of 10 μM for 24 h. Total cellular RNA was extracted for qRT-PCR analysis.

**RNA quantification.** Total RNA was isolated using the miRNeasy Mini Kit (QIAGEN, #217004) according to the manufacturer's instructions. RNA treated with DNase and the cDNAs were synthesized with PrimeScript™ RT reagent Kit with gDNA Eraser (TaKaRa, #RP047A). Samples without reverse transcriptase were also added as control. Gene expression was determined by amplifying the cDNA with ChamQ SYBR Color qPCR Master Mix(Vazyme, #Q431-02) by an ABI 7500 Fast Real-Time PCR System (Applied Biosystems). The expression of zebrafish *actb1* or EPC *actin* was used as an internal control. The $2^{-\Delta\Delta Ct}$ method was used to calculate relative expression changes. All qRT-PCR experiments were performed in triplicate and repeated three times. Primers used are listed in Supplementary Table 1.

**Preparation of recombinant protein.** The recombinant protein of Bmp8a has been successfully expressed in *Escherichia coli* and purified in our lab[37]. The recombinant protein of rStat1a or rStat1b was expressed and purified. Briefly, the recombinant plasmid (pET-28a-stat1a or pET-28a-stat1b) was transformed into *Trans*B (DE3) chemically competent cell. The positive transformants were incubated in liquid LB medium containing 50 μg/ml kanamycin at 37 °C by shaking at 160 rpm. When the cells grow to OD600 = 0.5, isopropyl β-D-thiogalactoside (IPTG) was added to a final concentration of 1 mM, and incubated at 19 °C with shaking at 120 rpm for 14 h. After incubation, the cells were harvested by centrifugation, re-suspended in PBS, and broken by ultrasound. The supernatant was centrifuged at $12,000 \times g$ for 20 min, and the protein was purified by Ni-NTA Sepharose column. The concentration of purified protein was determined by BCA (bicinchoninc acid) method. The purified recombinant proteins were stored at −80 °C for subsequent experiment.

**Electrophoretic mobility shift assay (EMSA).** EMSA was performed according to the instruction of Chemiluminescent EMSA Kit (Beyotime, #GS009). Oligonucleotides (Supplementary Table 1) were synthesized and biotinylated by Sangon Biotech Company as required and the complementary oligonucleotides were annealed to form the double DNA probe. The recombinant protein was pre-incubated in EMSA/Gel-Shift binding buffer with poly (dI-dC) at 25 °C for 25 min in the presence or absence of Non-Biotin probe (0.5 μM). Then, all samples were mixed with biotinylated probes (0.5 μM) and incubated at 25 °C for 30 min. The binding reaction without recombinant protein was set as the negative control. For supershift assay, anti-His tag antibody was incubated with the reaction mixture for another 30 min after the labeled probe was added. After separation in a native 6% polyacrylamide gel, the free probes and DNA-protein compound were transferred to a Hybond-N$^+$ nylon membrane, followed by crosslinking under UV light. After immersion in the blocking buffer, membranes were incubated with streptavidin conjugated HRP for 20 min with shaking and then washed three times with washing buffer. Membranes were exposed briefly in Western Lightning-ECL and then exposure to Automatic X-ray Film Processor (Smpicgg).

**Immunoblot analysis and co-immunoprecipitation.** For Immunoblot analysis, cells were lysed in NP-40 buffer (150 mM NaCl, 0.5% EDTA, 50 mM Tris, 1% NP40, proteinase inhibitors, and protein phosphatase inhibitors) for 30 min on ice. For co-immunoprecipitation (co-IP), whole-cell extracts were collected 48 h after transfection and lysed in NP-40 buffer on ice for 30 min. After centrifugation for 15 min at $12,000 \times g$, 4 °C, supernatants were collected and incubated with Protein A/G PLUS-Agarose (Santa Cruz, #sc-2003) coupled to specific antibodies for overnight with rotation at 4 °C. Beads were washed 3× times with NP-40 buffer. Bound proteins were eluted by boiling 7 min with 2× SDS sample buffer. For immunoblot analysis, immunoprecipitates or whole-cell lysates were separated by 12.5% SDS–PAGE gels, electro-transferred to PVDF membranes and blocked for 4 h with 5% no-fat milk solution, followed by blotting with the appropriate antibodies and detection by enhanced chemiluminiscence (ECL). Antibodies from Cell Signaling Technology: TBK1 (1:1000, #3504T), p-TBK1(Ser172) (1:1000, #5483T). Antibodies from Bioss: IRF3 (1:1000, #bs-2993R), p-IRF3 (Ser386) (1:1000, #bsm-52170R), p38MAPK (1:1000, #bs-0637R), p-p38MAPK (Thr180 + Tyr182) (1:1000, #bs-2210R), Actin (1:2000, #bs-0061R). Antibodies from Beyotime: HA tag (1:1000, #AH158) and Flag tag (1:1000, #AF519). Antibodies from CWBIO: His tag (1:5000, #CW0285) and goat antirabbit IgG HRP secondary antibody (1:8000, #CW0103S). Antibodies from Abcam: VeriBlot for IP Detection (1:5000, #ab131366).

**Statistics and reproducibility.** All statistical analysis was performed using Graphpad Prism 8.0.2. Data are presented as mean ± SD. Statistical significance between two groups was determined by two-tailed student's $t$-test. For comparison of multiple groups, the statistical analysis was performed using a one-way ANOVA followed by Games–Howell posthoc tests. Survival analyses were performed using the Kaplan–Meier method and assessed using the log-rank (Mantel-Cox) test. Representative experiments have been repeated at least two to three times. $*p < 0.05$, $**p < 0.01$, and $***p < 0.001$; ns not significant, $p > 0.05$.

**Reporting summary**. Further information on research design is available in the Nature Research Reporting Summary linked to this article.

## Data availability

All relevant data are available from the authors upon request and the corresponding author will be responsible for replying to the request. Source data underlying plots shown in figures are provided in Supplementary Data 1. Full blots are shown in Supplementary Fig. 4 of Supplementary Information.

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

## Acknowledgements

This work was supported by the grants of National Natural Science Foundation of China (31572259) and National Key Research and Development Project of the Ministry of Science and Technology (2018YFD0900505).

## Author contributions

Z.L.and S.-C.Z. conceived and coordinated the project. Z.L., G.J., and S.Z. designed the experiments. S.Z., H.L., and Y.-S.W. performed experiments. H.-Y.L. and Y.W. contributed to establish the *bmp8a* knockout mutant lines (*bmp8a*$^{-/-}$) using TALENs technology. S.Z. performed the statistical analysis. Z.L., S.-C.Z., and S.Z. wrote the manuscript, with input from the other authors. All authors reviewed the manuscript and approved the final version.

## Competing interests

The work is under a patent in China "A method to improve the antiviral immunity of fish (Application No. 201910454499.6)".
