## [Peer Review File · Communications Biology]

Reviewers' comments:

Reviewer #1 (Remarks to the Author):

Zhong et al. present a manuscript investigating the role of Bmp8a for antiviral immunity in zebrafish. Using a gain-of function and a loss-of function approach, the authors convincingly show that Bmp8a regulates resistance to viral infection in zebrafish. The authors also provide convincing data for the underlying regulatory pathway. The data is new and exciting. Bmp8a has not been associated to viral immunity before.

The manuscript reads very well and is easy to follow. In fact, it was a joy to read it. The provided data is presented in a clear and logic way and supports the claims of the authors. The scheme summarized the findings very well. I only have some minor critics, that could easily be addressed by the authors and recommend it for publication

1. I was wondering whether the authors also have some data on immunity against other classes of pathogens such as bacteria as this would even strengthen the findings.
2. I am not in favor of talking about „new players“ as this is not a football match. The title should be adapted. Also, please indicate that you worked on zebrafish in the title.
3. The authors use the students t test. For the comparison of more then two groups an ANOVA is more suitable. This should be corrected.
4. I33. What is „energy balance“?
5. I36. I think that the term morbidly is imprecise as this indicates disease. However, you have no read outs for disease course besides mortality. Therefore, this should be changed to mortality.
6. I118 It should be „death occurred“. As Mortality is the number of deaths in a certain cohort. OR the state of being mortal.
7. I 376: should be "Our study also shows". There are some more grammatical mistakes in the manuscript. The authors should revise it accordingly.

Reviewer #2 (Remarks to the Author):

The manuscript entitled „Bmp8a Is a Novel Player in Regulation of Antiviral Immunity“ by Zhong and colleagues investigated the role of bmp8 in innate immunity. They used bmp8a^{-/-} zebrafish infected with the viruses of GCRV, SVCV or TSVDV, and in vitro studies with cell lines. The results indicate that BMP8 is responsible for antiviral immunity, as bmp8 knockouts present with increased viral load and morbidity. IN vitro studies show an interaction of bmp8 with Alk6a. The signalling cascade of bmp8a causes phosphorylation of Tbk1 and Irf3 through p38 MAPK pathway. In addition, the IFN pathway is investigated in response to virus infection. Here the results show that, upon virus infection, bmp8a expression is activated through the binding of Stat1a/Stat1b to the GAS motifs.

The manuscript is excellently written, in perfect English without any typographical or stylistic error. Science is very well understandable, and conclusions can be supported by the data shown. So far, no other report on bmp8a and the role in antiviral immunity could be found.

Some concerns remain:

- Figure 3: The mRNA data shows strong effects- which the immunoblot (e.g. ALK6a protein do not support)
- Figure 5a and b- the effect of ALK6a is dominant. Can you explain how this effect can be supported? In mice, Alk6 mutations do not cause differences in iron in the liver. In addition, the expression level of Alk6 is much lower than of Alk2 or Alk3. Can you explain, how the expression level of zebrafish can be correlated to mice/humans?
- EPC and ZFL cells were grown in a 28 °C incubator supplied with 5% CO₂
The temperature seems to be very low to maintain cells.
- Fig 9, the graphical illustration is very well drawn.
- Students t test was statistically used: This implicates a normal distribution of values and comparison of two groups. Is this appropriate?
- The authors state that „Data were from three independent experiments (a–c, e) or two independent experiments“. I think you should have performed the experiments at least three times.
- Please state some words regarding the safety level applied to perform the virus experiments

Reviewer #3 (Remarks to the Author):

The study by Shenjie Zhong and colleagues presents a very interesting examination of the role of Bmp8a in antiviral immune responses in the zebrafish. The manuscript has several strengths, namely a combination of in vitro and in vivo work, looking at effects of over-expression, knock-down and knock-out of Bmp8a, including a few different viruses and cell lines, and investigating the mechanism of action of Bmp8a.

However there are a few important weaknesses that need addressing - general points and specific points.

General points

- 1) the claim of novelty. In the abstract, second sentence, it is stated "Whether BMP plays a role in antiviral immunity is unknown." This is wrong - please see Eddowes et al, Nature Microbiology 2019, Antiviral activity of bone morphogenetic proteins and activins. This paper is not cited in Zhong et al yet it clearly the single most relevant paper, both from a viewpoint of the overall concept, but also the molecular mechanism of action. The findings of Zhong et al need to be discussed in depth in the light of Eddowes et al and claims of novelty need to be toned down accordingly.
- 2) Role of SMADs. Bmps signal primarily through SMAD phosphorylation. SMAD phosphorylation was not measured in this paper (unless I missed it). This is important to act as a verification control for Bmp8a over expression and knock-down and knock-out; alternatives would be measuring mRNA expression of canonical BMP target genes (eg Id1). Small molecule inhibitors of smad phosphorylation were used in one experiment but these alone are insufficient because they have off-target effects. Again, to demonstrate they have been effective in the experiments in which they have been used, smad phosphorylation and/or target gene expression should be measured.
- 3) Use of recombinant Bmp8a protein. A lot of the data relies on over-expression of Bmp8a in cells, where the control is an empty vector. However, imposing a plasmid on a cell and enforcing over-expression of a particular protein can cause a stress response to a cell independently of the effects of the over-expressed protein itself. This stress response influences the metabolic state of the cell and could plausibly influence viral replication. Therefore a useful corroborative experiment to help the authors claims would be to test whether recombinant Bmp8a protein added to infected cells in culture exerts antiviral effects.
- 4) The rationale for choosing Bmp8a rather than other Bmps in zebrafish is very unclear. Were there

any screening experiments that tested other Bmps, but only Bmp8a had an antiviral effect?

Specific comments

a) Fig 1b and 1d- virus TCID assays were used to determine the viral titer in the supernatants, but there is no information in the methods about this assay and how the virus was quantified.

b) Fig 1e,f,g are striking, very impressive, but, does it say anywhere in the paper that the BMP8^{-/-} fish were left for 10+ days (no virus, no experimentation) to show that they are relatively happy? What if the KO did other things to the fish and resulted in lethality? BMPs are pleiotropic after all. Ideally this figure would have a third group of fish which were KO and not infected with a virus, and all of them would survive for 10+ days. Or, this info could be in the supps along with the KO method by TALEN.

c) At line 138, we are told that in zebrafish, there are four type I IFNs (of the subtype Phi). A citation is needed for this.

d) Supplementary figure 2 seems to imply that these authors, in the course of their BMP study, isolated a completely novel virus of turbot. I googled it to see if it was reported elsewhere and nothing came up, so I think I have understood them correctly when they say they found this new virus then used it in the study. Is this correct? If so, the sequence of the virus should be deposited.

e) Fig 4. Shouldn't be using t tests for these multiple comparisons. ANOVA should be used – even if the authors don't want to compare every group to every other group, there is a "many-to-one" dunnetts post-hoc test to compare to the control. I'm sure they're still statistically significant, but the correct test should be used.

f) As mentioned above in General comments, for this figure 4, it is important to have some westerns showing that when SMAD signalling was inhibited, that this could be seen by western blot at the dose of the drugs used by lack of pSMAD. Because this is quite an interesting finding that inhibiting SMAD 2/3 or SMAD1/5/8 signalling, the main canonical pathways, has no effect on the IFN induction, it should be confirmed that these pathways were indeed inhibited.

g) Fig 5k-n:

A dominant negative plasmid was used to impair expression of ALK6a, to assess the role ALK6a in expression of lots of type I IFN genes/ISGs/IRFs/TBK1. This looks convincing, but needs an extra control – a gene that ALK6a would not be expected to affect, to show that the use of this plasmid doesn't just downregulate lots of genes non-specifically, which happens to include type I IFN-related genes.

h) Fig 8a-c and e, again, should not be doing multiple t-tests here, even if it looks convincing and would probably be statistically significant by ANOVA. Should do that test instead with either Holm-Sidak all-to-all comparison or dunnetts all-to-one post-hocs.

i) Methods: RNAseq protocol: for each group, is it true that the authors took three fish, infected them etc, then harvested tissue for RNA extraction, and then pooled the three fish in each group, then took "three independent samples" from the pooled samples. For each group, the only thing independent about those samples would be that they went into three independent eppendorfs about 5 seconds apart. Or - were the samples from the three fish kept separate? Please clarify.

Point-by-Point Responses to the Reviewers' Critiques (COMMSBIO-20-2291-T)

We deeply appreciate the thorough analysis and constructive suggestions provided by the three reviewers to guide us to further improve our manuscript. As described in more detail below, we have addressed all the reviewers' concerns. With this extensive revision, we hope that the reviewers will concur with us that we have addressed all of the raised concerns in a satisfactory manner and, consequently, substantially strengthened our paper.

Reviewer #1 (Remarks to the Author):

Zhong *et al.* present a manuscript investigating the role of *Bmp8a* for antiviral immunity in zebrafish. Using a gain-of function and a loss-of function approach, the authors convincingly show that *Bmp8a* regulates resistance to viral infection in zebrafish. The authors also provide convincing data for the underlying regulatory pathway. The data is new and exciting. *Bmp8a* has not been associated to viral immunity before.

Response: We greatly appreciate you for your careful review and positive comments on our work.

1. I was wondering whether the authors also have some data on immunity against other classes of pathogens such as bacteria as this would even strengthen the findings.

Response: We sincerely appreciate the valuable and professional comments. Yes, we also did some studies on the anti-bacterial function of *Bmp8a*. For example, *Vibrio anguillarum* is an important pathogenic bacteria of aquaculture animals. We found that *bmp8a*^{-/-} zebrafish exhibited significantly reduced survival rate than wild-type zebrafish upon *Vibrio anguillarum* infection (Figure R1). This paper mainly focused on the antiviral function of *Bmp8a*, so the results of its anti-bacterial function were not presented.

Figure R1: Survival curve of WT or *bmp8a*^{-/-} zebrafish infection with live *Vibrio anguillarum*. Kaplan-Meier analysis of the overall survival of WT (n = 10) or *bmp8a*^{-/-} zebrafish (n = 10) which were injected i.p. with 20 μ l of live *Vibrio anguillarum* (1×10^8 CFU/ml) per fish. Data were analyzed by log-rank (Mantel-Cox) test (***) $p < 0.001$.

2. *I am not in favor of talking about „new players“ as this is not a football match. The title should be adapted. Also, please indicate that you worked on zebrafish in the title.*

Response: This is a good suggestion. As suggested, the title of this paper has been changed to “Bmp8a Is a Novel Essential Positive Regulator of Antiviral Immunity in Zebrafish”.

3. *The authors use the students t test. For the comparison of more than two groups an ANOVA is more suitable. This should be corrected.*

Response: Thanks for the comment and suggestion. In the revised manuscript, statistical analysis was performed using GraphPad Prism 8.0.2 software (La Jolla, CA). The alpha was set at 0.05. The distribution of data was assessed by the Shapiro–Wilk normality test. For comparison of multiple groups (Figure 2 q-s; Figure 3 s-u; Figure 4; Figure 5 c-e; Figure 6 i-k; Figure 8 a-c, e), the statistical analysis was performed using a one-way ANOVA followed by a Games-Howell Post Hoc multiple comparisons tests.

4. 33. *What is “energy balance“?*

Response: Energy always maintains a dynamic balance between intake and consumption, called energy balance. Positive energy balance leads to obesity in humans and other mammals in which excess energy is stored as triglycerides in adipose tissue. It has reported that BMP8B is a thermogenic protein that regulates energy balance in partnership with hypothalamic AMPK (Whittle et al. BMP8B increases brown adipose tissue thermogenesis through both central and peripheral actions. Cell 149, 871-885 (2012)). To make the meaning more clearly, the sentence was revised as “... and the regulation of adipogenesis and thermogenesis.” in the revised manuscript.

5. 36. *I think that the term morbidly is imprecise as this indicates disease. However, you have no read outs for disease course besides mortality. Therefore, this should be changed to mortality.*

Response: Thanks for the good suggestion. The word “morbidly” was changed to “mortality” in the revised manuscript.

6. 118. *It should be „death occurred“. As Mortality is the number of deaths in a certain cohort. OR the state of being mortal.*

Response: Thanks for the suggestion. The word “mortality” was changed to “death” in the revised manuscript.

7. 376. *should be “Our study also shows“. There are some more grammatical mistakes in the manuscript. The authors should revise it accordingly.*

Response: Thanks for the comments. We corrected the sentence according to the suggestion

in the revised manuscript. Moreover, we have carefully reviewed the manuscript and corrected the grammatical and typing errors as much as possible.

Reviewer #2 (Remarks to the Author):

The manuscript entitled „Bmp8a Is a Novel Player in Regulation of Antiviral Immunity“ by Zhong and colleagues investigated the role of bmp8 in innate immunity. They used bmp8a^{-/-} zebrafish infected with the viruses of GCRV, SVCV or TSV DV, and in vitro studies with cell lines. The results indicate that BMP8 is responsible for antiviral immunity, as bmp8 knockouts present with increased viral load and morbidity. IN vitro studies show an interaction of bmp8 with Alk6a. The signalling cascade of bmp8a causes phosphorylation of Tbk1 and Irf3 through p38 MAPK pathway. In addition, the IFN pathway is investigated in response to virus infection. Here the results show that, upon virus infection, bmp8a expression is activated through the binding of Stat1a/Stat1b to the GAS motifs.

Response: We thank the reviewer for the encouraging words and the appreciation of our study. We also thank the reviewer for some critical observations addressed below and her/his important suggestions.

1. Figure 3: The mRNA data shows strong effects- which the immunoblot (e.g. ALk6a protein do not support).

Response: Oh, we did not detected the expression of the Alk6a protein in Figure 3. In our view, the mRNA data here are generally in line with the immunoblot data in Figure 3.

2. Figure 5a and b- the effect of ALK6a is dominant. Can you explain how this effect can be supported? In mice, Alk6 mutations do not cause differences in iron in the liver. In addition, the expression level of Alk6 is much lower than of Alk2 or Alk3. Can you explain, how the expression level of zebrafish can be correlated to mice/humans?

Response: We thank the reviewer for this comment which indeed is an intriguing question. Right now, it does not know which BMP receptor(s) is (are) involved in the regulation of the antiviral immune responses. Our data (Figure 5a, b) showed that, among the receptors (*alk2*, *alk3*, *alk6a*, *bmpr2a*, *bmpr2b*, *actr2a*, *actr2b*), *alk6a* has the highest expression upon poly(I:C) or virus challenge. Our further experiments (Figure 5c-n) support that Alk6a was involved in the antiviral signaling (Tbk1-Irf3/7-Ifn). In Figure 5a and b, the expression levels of these BMP receptor in the control group were all set as 1. Their true expression levels were restored as following (Figure R2). The data in Figure R2 reveal that the expression level of Alk6 in ZFL cells is also much lower than that of Alk2 or Alk3.

Figure R2: Expression of *alk2*, *alk3*, *alk6a*, *bmpr2a*, *bmpr2b*, *actr2a*, *actr2b* mRNA in ZFL cells stimulated with poly(I:C) (a) or GCRV (b) for 48 h. Data were from three independent experiments and were analyzed by Student's *t*-test (two-tailed). All data were presented as mean \pm SD (**p* < 0.05, ***p* < 0.01, ****p* < 0.001, ns means no significant difference).

3. EPC and ZFL cells were grown in a 28 °C incubator supplied with 5% CO₂. The temperature seems to be very low to maintain cells.

Response: Regarding the culture conditions (Atmosphere: air, 95%; carbon dioxide (CO₂), 5%; Temperature: 28°C) of these two strains of cells, we strictly follow the operating specifications on CZRC. These two cells are in excellent condition under this culture method. We are sure that the temperature is appropriate to maintain EPC and ZFL cells.

4. Fig 9, the graphical illustration is very well drawn.

Response: Thank you.

5. Student's *t* test was statistically used: This implicates a normal distribution of values and comparison of two groups. Is this appropriate?

Response: Thanks for the comments. In the revised manuscript, statistical analysis was performed using GraphPad Prism 8.0.2 software (La Jolla, CA). The alpha was set at 0.05. The distribution of data was assessed by the Shapiro–Wilk normality test. For comparison of multiple groups (Figure 2 q-s; Figure 3 s-u; Figure 4; Figure 5 c-e; Figure 6 i-k; Figure 8 a-c, e), the statistical analysis was performed using a one-way ANOVA followed by a Games-Howell Post Hoc multiple comparisons tests.

6. The authors state that “ Data were from three independent experiments (a–c, e) or two independent experiments”. I think you should have performed the experiments at least three times.

Response: Thanks. We totally agree with the reviewer’s opinion that we should have performed the experiments at least three times. In fact, most of the experiments in this paper have been repeated 3 times. Only the experiments of Figure 1f-h and Figure 8f-k were repeated twice. Although the experiments of Figure 1f-h were repeated twice, three different viruses were tested and the results are similar, indicating the conclusion is strong. For Figure 8f-k, we repeated these experiments, and the results were consistent with the previous data. We added the information in the revised manuscript.

7. Please state some words regarding the safety level applied to perform the virus experiments.

Response: Thanks. All experiments related to the virus were conducted in the biosafety cabinet (BL-2 level). The mammals are not susceptible to these viruses. The explanation has been now included in the revised manuscript.

Reviewer #3 (Remarks to the Author):

The study by Shenjie Zhong and colleagues presents a very interesting examination of the role of Bmp8a in antiviral immune responses in the zebrafish. The manuscript has several strengths, namely a combination of in vitro and in vivo work, looking at effects of over-expression, knock-down and knock-out of Bmp8a, including a few different viruses and cell lines, and investigating the mechanism of action of Bmp8a.

However there are a few important weaknesses that need addressing - general points and specific points.

Response: We thank this reviewer for appreciating our study and providing insightful comments.

General points

1. *the claim of novelty. In the abstract, second sentence, it is stated "Whether BMP plays a role in antiviral immunity is unknown." This is wrong - please see Eddowes et al, Nature Microbiology 2019, Antiviral activity of bone morphogenetic proteins and activins. This paper is not cited in Zhong et al yet it clearly the single most relevant paper, both from a viewpoint of the overall concept, but also the molecular mechanism of action. The findings of Zhong et al need to be discussed in depth in the light of Eddowes et al and claims of novelty need to be toned down accordingly.*

Response: We thank the reviewer for this important observation. The claim of novelty of our

finding was modified as “However, the understanding of role of BMPs in antiviral immunity is still limited”. Also, more introduction and discussion regarding the work of Eddowes et al. in Nature Microbiology have been included in the revised manuscript.

2. Role of SMADs. *Bmps signal primarily through SMAD phosphorylation. SMAD phosphorylation was not measured in this paper (unless I missed it). This is important to act as a verification control for Bmp8a over expression and knock-down and knock-out; alternatives would be measuring mRNA expression of canonical BMP target genes (eg Id1). Small molecule inhibitors of smad phosphorylation were used in one experiment but these alone are insufficient because they have off-target effects. Again, to demonstrate they have been effective in the experiments in which they have been used, smad phosphorylation and/or target gene expression should be measured.*

Response: We sincerely appreciate the valuable and professional comments. We performed some experiments mentioned here. Overexpression of *bmp8a* increased the phosphorylation of SMAD1/5/8 or SMAD2/3 in ZFL (Figure R3a) or EPC (Figure R3b) cells. However, the phosphorylation of SMAD1/5/8 or SMAD2/3 was inhibited in the *bmp8a*-knockdown ZFL (Figure R3c) cells, as well as in the *bmp8a*^{-/-} zebrafish liver tissue (Figure R3d). We also confirmed that DMH1 could inhibit the SMAD1/5/8 phosphorylation in ZFL (Figure R3e) or EPC (Figure R3g) cells, and TP0427736 HCl could inhibit the SMAD2/3 phosphorylation in ZFL (Figure R3f) or EPC (Figure R3h) cells.

As a positive control example, we also checked the canonical BMP target gene *id1* mRNA expression (Figure R3i-m). Obviously, overexpression of *bmp8a* increased the *id1* mRNA expression (Figure R3i), while *id1* mRNA expression was downregulated in the *bmp8a*-knockdown ZFL (Figure R3j) cells or in the *bmp8a*^{-/-} zebrafish liver tissue (Figure R3k). Also, the SMAD1/5/8 inhibitor DMH1 could decrease the *id1* mRNA expression (Figure R3m).

Figure R3: Bmp8a increased the SMAD phosphorylation. a, b Immunoblot analysis of phosphorylated (p-) smad1/5/8 and (p-) smad2/3 after transfected with 2 μ g *bmp8a* or empty vector in ZFL (a) or EPC (b) cells. The cells were collected at 36 h or 48 h post transfection. c Immunoblot analysis of phosphorylated (p-) smad1/5/8 and (p-) smad 2/3 after *bmp8a* knockdown in ZFL cells. The cells were collected at 24 h and 36 h post knockdown. d Immunoblot analysis of phosphorylated (p-) smad1/5/8 and (p-) smad 2/3 in WT and *bmp8a*^{-/-} zebrafish liver tissue. e, g Immunoblot analysis of phosphorylated (p-) smad1/5/8 after treated with DMH1 (at concentrations of 10 μ M) in ZFL(e) or EPC (g) cells. The cells were collected at 24 h post treated. f, h Immunoblot analysis of phosphorylated (p-) smad2/3 after treated with TP0427736 HCl (at concentrations of 10 μ M) in ZFL(f) or EPC (h) cells. The cells were collected at 24 h post treated. i Expression of *id1* mRNA after transfected with *bmp8a* (2 μ g) or empty vector (2 μ g) in ZFL cells. The cells were collected at 36 h or 48 h post transfection. j Expression of *id1* mRNA after knockdown *bmp8a* in ZFL cells. The cells were collected at 24 h or 36 h post transfection. k The expression of *id1* mRNA in the liver tissues from wild-type (WT) or *bmp8a*^{-/-} zebrafish. l The expression of *id1* mRNA after treated with DMH1 (at concentrations of 10 μ M) in ZFL cells. The cells were collected at 24 h post treated. Data were from three independent experiments and were analyzed by Student's *t*-test (two-tailed). All data were presented as mean \pm SD (***p* < 0.01, ****p* < 0.001).

3. Use of recombinant Bmp8a protein. A lot of the data relies on over-expression of Bmp8a in cells, where the control is an empty vector. However, imposing a plasmid on a cell and enforcing over-expression of a particular protein can cause a stress response to a cell independently of the effects of the over-expressed protein itself. This stress response influences the metabolic

state of the cell and could plausibly influence viral replication. Therefore a useful corroborative experiment to help the authors claims would be to test whether recombinant Bmp8a protein added to infected cells in culture exerts antiviral effects.

Response: We sincerely appreciate the valuable and professional comments. Indeed, the recombinant protein of Bmp8a (rBmp8a) has been successfully expressed in *Escherichia coli* and purified in our lab (Zhong et al. Spatial and temporal expression of bmp8a and its role in regulation of lipid metabolism in zebrafish *Danio rerio*. Gene Rep. 10, 33-41 (2018)). We added the experiments to test whether rBmp8a protein exerts antiviral effects in the ZFL (Figure R4a, b) or EPC (Figure R4c, d) cells. It suggests that Bmp8a protein inhibits RNA viral replication, which is consistent with the previous data relied on the over-expression of Bmp8a in cells. The data were added in the revised manuscript.

Figure R4: rBmp8a inhibits RNA viral replication. a, b ZFL cells treated with rBmp8a (with final concentrations of 5 $\mu\text{g/ml}$) or PBS co-incubation with GCRV ($5 \times 10^4 \text{TCID}_{50}$ per ml), and the culture supernatants were collected at 72 h post-infection. The cell monolayers were fixed with 4% PFA for 1 h and stained with 0.5% crystal violet for 2 h (a), and the viral titers of the supernatants were determined by TCID_{50} assays (b). c, d Similar as (a, b) but in EPC cells. Data were from three independent experiments and were analyzed by Student's *t*-test (two-tailed). All data were presented as mean \pm SD (** $p < 0.01$, *** $p < 0.001$, ns means no significant difference).

4. The rationale for choosing Bmp8a rather than other Bmps in zebrafish is very unclear. Were there any screening experiments that tested other Bmps, but only Bmp8a had an antiviral effect?

Response: Thank you very much for the important comments. This is a good suggestion to perform screening experiments to test other BMPs, but we did not do this currently. So it still does not know exactly whether other BMPs have anti-viral function in zebrafish. We created *bmp8a* knock out zebrafish in our lab. Then we we carried out transcriptome analysis of the livers of *bmp8a*^{-/-} mutant and wild-type zebrafish. We found that the down-regulated genes in the mutant were remarkably enriched in the the antiviral immune process, so we tested its antiviral function. Then we have this story.

Specific comments

a). Fig 1b and 1d- virus TCID assays were used to determine the viral titer in the supernatants, but there is no information in the methods about this assay and how the virus was quantified.

Response: The reviewer raises an important issue and of great concern. The viral titer was determined by a 50% tissue culture-infective dose (TCID₅₀) assay on EPC cells according to the method of Reed and Muench. We added the detail information in the revised manuscript.

b). Fig 1e,f,g are striking, very impressive, but, does it say anywhere in the paper that the BMP8^{-/-} fish were left for 10+ days (no virus, no experimentation) to show that they are relatively happy? What if the KO did other things to the fish and resulted in lethality? BMPs are pleiotropic after all. Ideally this figure would have a third group of fish which were KO and not infected with a virus, and all of them would survive for 10+ days. Or, this info could be in the supps along with the KO method by TALEN.

Response: Thank you very much for raising this interesting issue. The zebrafish *bmp8a* knockout mutant lines (*bmp8a*^{-/-}) can survive and breed normally under standard laboratory conditions. In fact, the mutant lines (*bmp8a*^{-/-}) have been bred for several generations. The information has been added in the revised manuscript.

c). At line 138, we are told that in zebrafish, there are four type I IFNs (of the subtype Phi). A citation is needed for this.

Response: We sincerely appreciate the professional comments. The reference was cited in the revised manuscript.

d). Supplementary figure 2 seems to imply that these authors, in the course of their BMP study, isolated a completely novel virus of turbot. I googled it to see if it was reported elsewhere and nothing came up, so I think I have understood them correctly when they say they found this new virus then used it in the study. Is this correct? If so, the sequence of the virus should be deposited.

Response: Yes, you are right. TSVDV is a completely novel virus of turbot. The virus was first described by Qin et al. (2008, in Chinese with English abstract). It was confirmed to be a virus under electron microscope, but the sequence of the virus is unknown. In our study, we isolated this virus from the focal site of turbot skin. We are also very interested to sequence the genome of this virus in the future.

e) Fig 4. Shouldn't be using *t* tests for these multiple comparisons. ANOVA should be used – even if the authors don't want to compare every group to every other group, there is a “many-to-one” dunnetts post-hoc test to compare to the control. I'm sure they're still statistically significant, but the correct test should be used.

Response: We apologize for these deficiencies in the original manuscript. In the revised

manuscript, statistical analysis was performed using GraphPad Prism 8.0.2 software (La Jolla, CA). The alpha was set at 0.05. The distribution of data was assessed by the Shapiro–Wilk normality test. For comparison of multiple groups (Figure 2 q-s; Figure 3 s-u; Figure 4; Figure 5 c-e; Figure 6 i-k; Figure 8 a-c, e), the statistical analysis was performed using a one-way ANOVA followed by a Games-Howell Post Hoc multiple comparisons tests.

f) As mentioned above in General comments, for this figure 4, it is important to have some westerns showing that when SMAD signalling was inhibited, that this could be seen by western blot at the dose of the drugs used by lack of pSMAD. Because this is quite an interesting finding that inhibiting SMAD 2/3 or SMAD1/5/8 signalling, the main canonical pathways, has no effect on the IFN induction, it should be confirmed that these pathways were indeed inhibited.

Response: We sincerely appreciate the valuable and professional comments. Same to the response to the General comments above, we confirmed that the inhibitor DMH1 could inhibit the SMAD1/5/8 phosphorylation in ZFL (Figure R3e) or EPC (Figure R3g) cells, and the inhibitor TP0427736 HCl could inhibit the SMAD2/3 phosphorylation in ZFL (Figure R3f) or EPC (Figure R3h) cells. Therefore, it was confirmed that these pathways were indeed inhibited.

) Fig 5k-n: A dominant negative plasmid was used to impair expression of ALK6a, to assess the role ALK6a in expression of lots of type I IFN genes/ISGs/IRFs/TBK1. This looks convincing, but needs an extra control – a gene that ALK6a would not be expected to affect, to show that the use of this plasmid doesn't just downregulate lots of genes non-specifically, which happens to include type I IFN-related genes.

Response: We sincerely appreciate the valuable and professional comments. The gene of GAPDH (glyceraldehyde 3-phosphate dehydrogenase) was selected as the extra control. GAPDH is an enzyme in the process of glycolysis. The gene encoding this enzyme is a housekeeper gene, and it is commonly used as a standardized internal reference for molecular biology experiments related to protein, RNA, DNA, etc. The expression of GAPDH would not be expected to be affected by Alk6a. Our experiments confirmed this (Figure R5).

Figure R5: The *alk6a*-ΔGS plasmid can not affect the expression of *gadph*. **a, c** Expression of *gadph* mRNA after transfected with 2 μg pcDNA3.1-*alk6a*-ΔGS or empty vector in ZFL (**a**) or EPC (**c**) cells for 48 h. **b, d** Expression of *gadph* mRNA after transfected with 2 μg pcDNA3.1-*alk6a*-ΔGS or empty vector in ZFL (**b**) or EPC (**d**) cells for 24 h, followed by infection with GCRV for another 36 h. The expression of zebrafish *actb1* or EPC *actin* was used as an internal control for the qRT-PCR. Data were from three independent experiments and were analyzed by Student's *t*-test (two-tailed) and were presented as mean ± SD (ns means no significant difference).

h) Fig 8a-c and e, again, should not be doing multiple t-tests here, even if it looks convincing and would probably be statistically significant by ANOVA. Should do that test instead with either Holm-Sidak all-to-all comparison or dunnetts all-to-one post-hocs.

Response: We apologize for these deficiencies in the original manuscript. In the revised manuscript, statistical analysis was performed using GraphPad Prism 8.0.2 software (La Jolla, CA). The alpha was set at 0.05. The distribution of data was assessed by the Shapiro–Wilk normality test. For comparison of multiple groups (Figure 2 q-s; Figure 3 s-u; Figure 4; Figure 5 c-e; Figure 6 i-k; Figure 8 a-c, e), the statistical analysis was performed using a one-way ANOVA followed by a Games-Howell Post Hoc multiple comparisons tests.

i) Methods: RNAseq protocol: for each group, is it true that the authors took three fish, infected them etc, then harvested tissue for RNA extraction, and then pooled the three fish in each group, then took “three independent samples” from the pooled samples. For each group, the only thing independent about those samples would be that they went into three independent eppendorfs about 5 seconds apart. Or - were the samples from the three fish kept separate? Please clarify.

Response: We sincerely appreciate the valuable and professional comments. We regret that our description may be ambiguous. In fact, nine adult zebrafish ($0.4 \text{ g} \pm 0.05 \text{ g}$) from each group (wild type group and *bmp8a*^{-/-} group) were randomly selected and were injected

intraperitoneally with 50 μ l of GCRV (10^8 TCID₅₀ per ml) per fish. Seventy-two hours after the injection, the livers from three fish in the same group were pooled together for total RNA extraction, which serve as an independent sample. Therefore, three independent samples for each group were prepared for transcriptome sequencing. The detail information has been included in the revised manuscript.

REVIEWERS' COMMENTS:

Reviewer #1 (Remarks to the Author):

The authors have answered all my inquiries. I recommend publication.

Reviewer #2 (Remarks to the Author):

The authors have responded to my comments.

Reviewer #3 (Remarks to the Author):

The authors have adequately addressed the criticisms raised.

Point-by-Point Responses to the Reviewers' Critiques (COMMSBIO-20-2291-T)

We deeply appreciate the constructive suggestions and kind comment to publish this study. As described in more detail below, we have addressed all the reviewers' concerns.

Reviewer #1 (Remarks to the Author):

The authors have answered all my inquiries. I recommend publication.

Response: We thank the reviewer for the kind comments on our revised manuscript.

Reviewer #2 (Remarks to the Author):

The authors have responded to my comments.

Response: We thank the reviewer for the kind comments on our revised manuscript.

Reviewer #3 (Remarks to the Author):

The authors have adequately addressed the criticisms raised.

Response: We thank the reviewer for the kind comments on our revised manuscript.